# Extracting grid cell characteristics from place cell inputs using non-negative principal component analysis

**Yedidyah Dordek[1,2†], Daniel Soudry[3,4*†], Ron Meir[1], Dori Derdikman[2*]**

[1]Faculty of Electrical Engineering, Technion – Israel Institute of Technology, Haifa, Israel; [2]Rappaport Faculty of Medicine and Research Institute, Technion – Israel Institute of Technology, Haifa, Israel; [3]Department of Statistics, Columbia University, New York, United States; [4]Center for Theoretical Neuroscience, Columbia University, New York, United States

**Abstract** Many recent models study the downstream projection from grid cells to place cells, while recent data have pointed out the importance of the feedback projection. We thus asked how grid cells are affected by the nature of the input from the place cells. We propose a single-layer neural network with feedforward weights connecting place-like input cells to grid cell outputs. Place-to-grid weights are learned via a generalized Hebbian rule. The architecture of this network highly resembles neural networks used to perform Principal Component Analysis (PCA). Both numerical results and analytic considerations indicate that if the components of the feedforward neural network are non-negative, the output converges to a hexagonal lattice. Without the non-negativity constraint, the output converges to a square lattice. Consistent with experiments, grid spacing ratio between the first two consecutive modules is $-1.4$. Our results express a possible linkage between place cell to grid cell interactions and PCA.

*For correspondence: daniel. soudry@gmail.com (DS); derdik@ technion.ac.il (DD)

[†]These authors contributed equally to this work

**Competing interests:** The authors declare that no competing interests exist.

## Introduction

The system of spatial navigation in the brain has recently received much attention (**Burgess, 2014**; **Morris, 2015**; **Eichenbaum, 2015**). This system involves many regions, which seem to divide into two major classes: regions such as CA1 and CA3 of the hippocampus, which contain place cells (**O'Keefe and Dostrovsky, 1971**; **O'Keefe and Nadel, 1978**), vs. regions, such as the medial-entorhinal cortex (MEC), the presubiculum and the parasubiculum, which contain grid cells, head-direction cells and border cells (**Hafting et al., 2005**; **Boccara et al., 2010**; **Sargolini et al., 2006**; **Solstad et al., 2008**; **Savelli et al., 2008**). While the phenomenology of those cells is described in many studies (**Derdikman and Knierim, 2014**; **Tocker et al., 2015**), the manner in which grid cells are formed is quite enigmatic. Many mechanisms have been proposed. The details of these mechanisms differ, however, they mostly share in common the assumption that the animal's velocity is the main input to the system (**Derdikman and Knierim, 2014**; **Zilli, 2012**; **Giocomo et al., 2011**), such that positional information is generated by the integration of this input in time. This process is termed 'path integration' (PI) (**Mittelstaedt and Mittelstaedt, 1980**). A notable exception to this class of models was suggested in a previous paper by **Kropff and Treves (2008)**; and in a sequel to that paper (**Si and Treves, 2013**), in which they demonstrated the emergence of grid cells from place cell inputs without using the rat's velocity as an input signal.

We note here that generating grid cells from place cells may seem at odds with the architecture of the network, since it is known that place cells reside at least one synapse downstream of grid cells (**Witter and Amaral, 2004**). Nonetheless, there is current evidence that the feedback from place

**eLife digest** Long before the invention of GPS systems, ships used a technique called dead reckoning to navigate at sea. By tracking the ship's speed and direction of movement away from a starting point, the crew could estimate their position at any given time. Many believe that some animals, including rats and humans, can use a similar process to navigate in the absence of external landmarks. This process is referred to as "path integration".

It is commonly believed that the brain's navigation system is based on such path integration in two key regions: the entorhinal cortex and the hippocampus. Most models of navigation assume that a network of grid cells in the entorhinal cortex processes information about an animal's speed and direction of movement. The grid cell network estimates the animal's future position and relays this information to cells in the hippocampus called place cells. Individual place cells then fire whenever the animal reaches a specific location.

However, recent work has shown that information also flows from place cells back to grid cells. Further experiments have suggested that place cells develop before grid cells. Also, inactivating place cells eliminates the hexagonal patterns that normally appear in the activity of the grid cells.

Using a computational model, Dordek, Soudry et al. now show that place cell activity could in principle trigger the formation of the grid cell network, rather than vice versa. This is achieved using a process that resembles a common statistical algorithm called principal component analysis (PCA). However, this only works if place cells only excite grid cells and never inhibit their activity, similar to what is known from the anatomy of these brain regions. Under these circumstances, the model shows hexagonal patterns emerging in the activity of the grid cells, with similar properties to those patterns observed experimentally.

These results suggest that navigation may not depend solely on grid cells processing information about speed and direction of movement, as assumed by path integration models. Instead grid cells may rely on position-based input from place cells. The next step is to create a single model that combines the flow of information from place cells to grid cells and vice versa.

cells to grid cells is of great functional importance. Specifically, there is evidence that inactivation of place cells causes grid cells to disappear (*Bonnevie et al., 2013*), and furthermore, it seems that, in development, place cells emerge before grid cells do (*Langston et al., 2010*; *Wills et al., 2010*). Thus, there is good motivation for trying to understand how the feedback from hippocampal place cells may contribute to grid cell formation.

In the present paper, we thus investigated a model of grid cell development from place cell inputs. We showed the resemblance between a feedforward network from place cells to grid cells to a neural network architecture previously used to implement the PCA algorithm (*Oja, 1982*). We demonstrated, both analytically and through simulations, that the formation of grid cells from place cells using such a neural network could occur given specific assumptions on the input (i.e. zero mean) and on the nature of the feedforward connections (specifically, non-negative, or excitatory).

## Results

### Comparing neural-network results to PCA

We initially considered the output of a single-layer neural network and of the PCA algorithm in response to the same inputs. These consisted of the temporal activity of a simulated agent moving around in a two-dimensional (2D) space (*Figure 1A*; see Materials and methods for details). In order to mimic place cell activity, the simulated virtual space was covered by multiple 2D Gaussian functions uniformly distributed at random (*Figure 1B*), which constituted the input. In order to calculate the principal components, we used a [Neuron x Time] matrix (*Figure 1C*) after subtracting the temporal mean, generated from the trajectory of the agent as it moved through the place fields. Thus, we displayed a one-dimensional mapping of the two-dimensional activity, transforming the 2D activity into a 1D vector per input neuron. This resulted in the [Neuron X Neuron] covariance matrix (*Figure 1D*), on which PCA was performed by evaluating the appropriate eigenvalues and eigenvectors.

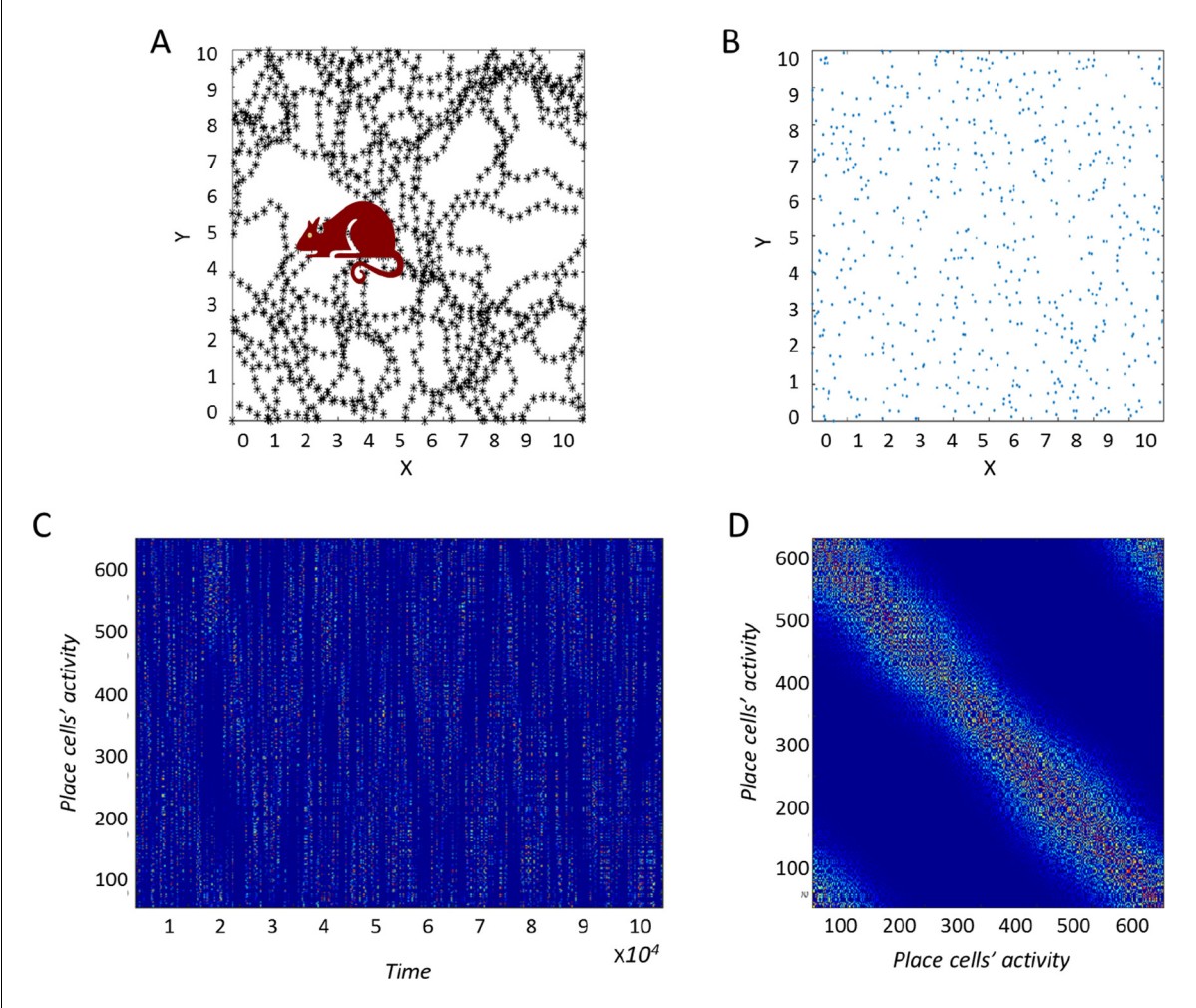

**Figure 1.** Construction of the correlation matrix from behavior. (A) Diagram of the environment. Black dots indicate places the virtual agent has visited. (B) Centers of place cells uniformly distributed in the environment. (C) The [Neuron X Time] matrix of the input-place cells. (D) Correlation matrix of (C) used for the PCA process.

To learn the grid cells, based on the place cell inputs, we implemented a single-layer neural network with a single output (*Figure 2*). Input to output weights were governed by a Hebbian-like learning rule. As described in the Introduction (see also analytical treatment in the Methods section), this type of architecture induces the output's weights to converge to the leading principal component of the input data.

The agent explored the environment for a sufficiently long time allowing the weights to converge to the first principal component of the temporal input data. In order to establish a spatial interpretation of the eigenvectors (from PCA) or the weights (from the converged network) we projected both the PCA eigenvectors and the network weights onto the place cells space, producing corresponding spatial activity maps. The leading eigenvectors of the PCA and the network's weights converged to square-like periodic spatial solutions (*Figure 3A–B*).

Being a PCA algorithm, the spatial projections of the weights were periodic in space due to the covariance matrix of the input having a Toeplitz structure (*Dai et al., 2009*) (a Toeplitz matrix has constant elements along each diagonal). Intuitively, the Toeplitz structure arises due to the spatial stationarity of the input. In fact, since we used periodic boundary conditions for the agent's motion, the covariance matrix was a circulant matrix, and the eigenvectors were sinusoidal functions, with length constants determined by the scale of the box (*Gray, 2006*) [a circulant matrix is defined by a single row (or column), and the remaining rows (or columns) are obtained by cyclic permutations. It

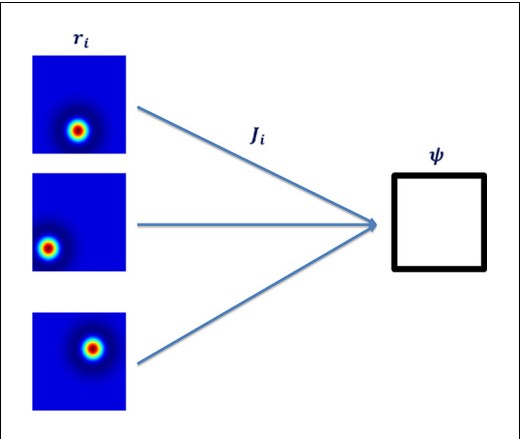

**Figure 2.** Neural network architecture with feedforward connectivity. The input layer corresponds to place cells and the output to a single cell.

is a special case of a Toeplitz matrix - see for example *Figure 1D*]. The covariance matrix was heavily degenerate, with approximately 90% of the variance accounted for by the first 15% of the eigenvectors (*Figure 4B*). The solution demonstrated a fourfold redundancy. This was apparent in the plotted eigenvalues (from the largest to the smallest eigenvalue, *Figure 4A and C*), which demonstrated a fourfold grouping-pattern. The fourfold redundancy can be explained analytically by the symmetries of the system – see analytical treatment of PCA in Methods section (specifically Figure 15C).

In summary, both the direct PCA algorithm and the neural network solutions developed periodic structure. However, this periodic structure was not hexagonal but rather had a square-like form.

## Adding a non-negativity constraint to the PCA

It is known that most synapses from the hippocampus to the MEC are excitatory (*Witter and Amaral, 2004*). We thus investigated how a non-negativity constraint, applied to the projections from place cells to grid cells, affected our simulations. As demonstrated in the analytical treatment in the

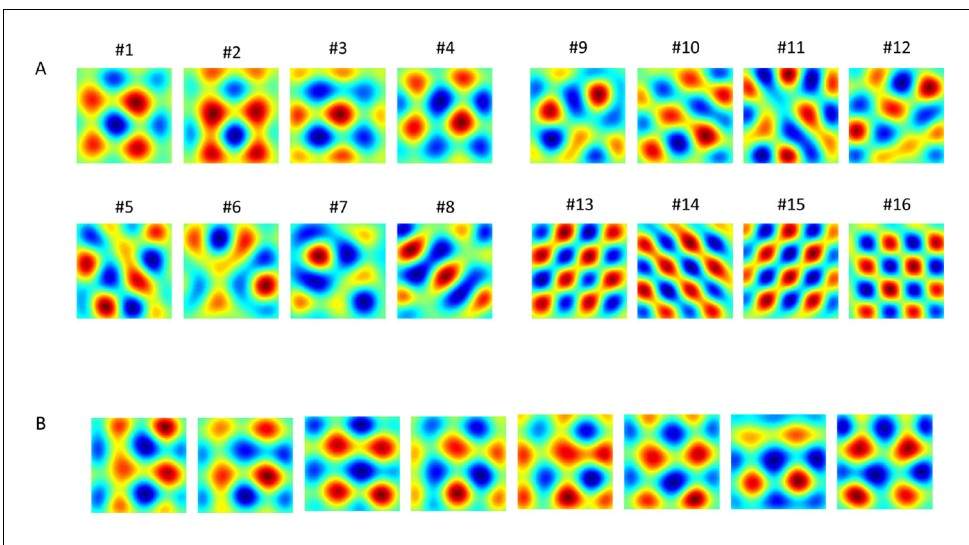

**Figure 3.** Results of PCA and of the networks' output (in different simulations). (**A**) 1st 16 PCA eigenvectors projected on the place cells' input space. (**B**) Converged weights of the network (each result from different simulation, initial conditions and trajectory) projected onto place cells' space. Note that the 8 outputs shown here resemble linear combinations of components #1 to #4 in panel A.

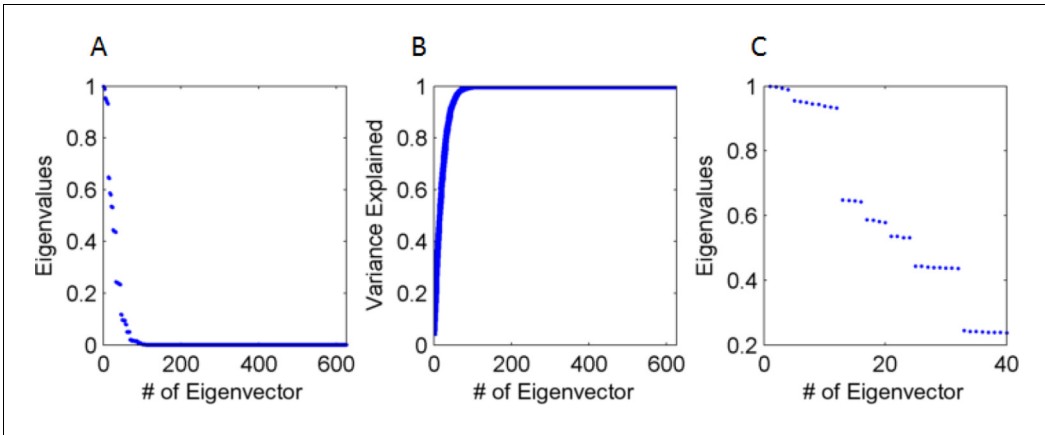

**Figure 4.** Eigenvalues and eigenvectors of the input's correlation matrix. (**A**) Eigenvalue size (normalized by the largest, from large to small (**B**) Cumulative explained variance by the eigenvalues, with 90% of variance accounted for by the first 35 eigenvectors (out of 625). (**C**) Amplitude of leading 32 eigenvalues, demonstrating that they cluster in groups of 4 or 8. Specifically, the first four clustered groups correspond respectively (from high to low) to groups A,B,C & D In *Figure 15C*, which have the same redundancy (4,8,4 & 4).

Methods section, we could expect to find hexagons when imposing the non-negativity constraint. Indeed, when adding this constraint, the outputs behaved in a different manner and converged to a **hexagonal grid**, similar to real grid cells. While it was straightforward to constrain the neural network, calculating non-negative PCA directly was a more complicated task due to the non-convex nature of the problem (*Montanari and Richard, 2014*; *Kushner and Clark, 1978*).

In the network domain, we used a simple rectification rule for the learned feedforward weights, which constrained their values to be non-negative. For the direct non-negative PCA calculation, we used the raw place cells activity (after spatial or temporal mean normalization), as inputs to three different iterative numerical methods: NSPCA (Nonnegative Sparse PCA), AMP (Approximate Message Passing) and FISTA (Fast Iterative Threshold and Shrinkage) based algorithms (see Materials and methods section).

In both cases, we found that hexagonal grid cells emerged in the output layer (plotted as spatial projection of weights and eigenvectors: *Figure 5A–B*, *Figure 6A–B*, *Video 1*, *Video 2*). When we repeated the process over many simulations (i.e. new trajectories and random initializations of weights) we found that the population as a whole consistently converged to hexagonal grid-like responses, while similar simulations with the unconstrained version did not (compare *Figure 3* to *Figure 5–Figure 6*).

In order to further assess the hexagonal grid emerging in the output, we calculated the mean (hexagonal) Gridness scores ([*Sargolini et al., 2006*], which measure the degree to which the solution resembles a hexagonal grid [see Materials and methods]). We ran about 1500 simulations of the network (in each simulation, the network consisted of 625 place cell-like inputs and a single grid cell-like output), and found noticeable differences between the constrained and unconstrained cases. Namely, the Gridness score in the non-negatively constrained-weight simulations was significantly higher than in the unconstrained-weight case (Gridness = 1.07 ± 0.003 in the constrained case vs. 0.302 ± 0.003 in the unconstrained case. see *Figure 7*). A similar difference was observed with the direct non-negative PCA methods (1500 simulations, each with different trajectories, Gridness = 1.13 ± 0.0022 in the constrained case vs. 0.27 ± 0.0023 in the unconstrained case).

Another score we tested was a 'Square Gridness' score (see Materials and methods) where we measured the 'Squareness' of the solutions (as opposed to 'Hexagonality'). We found that the unconstrained network had a higher square-Gridness score while the constrained network had a lower square-Gridness score (*Figure 7*); for both the direct-PCA calculation (square-Gridness = 0.89 ± 0.0074 in the unconstrained case vs. 0.1 ± 0.006 in the constrained case) and the neural-network (square-Gridness = 0.073 ± 0.006 in the constrained case vs. 0.73 ± 0.008 in the unconstrained case).

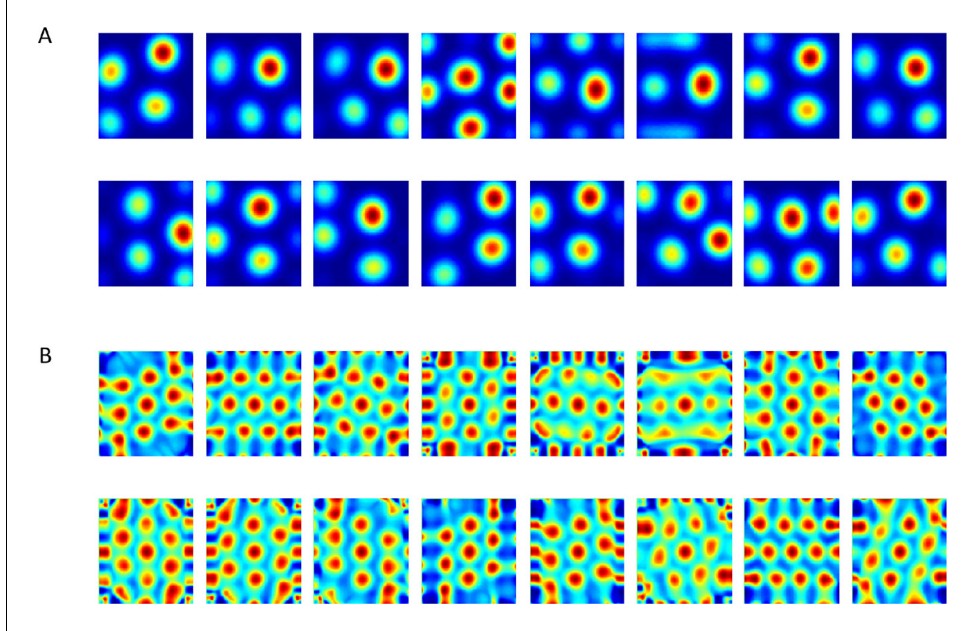

**Figure 5.** Output of the neural network when weights are constrained to be non-negative. (**A**) Converged weights (from different simulations) of the network projected onto place cells space. See an example of a simulation in *Video 1*. (**B**) Spatial autocorrelations of (**A**). See an example of the evolution of autorcorrelation in simulation in *Video 2*.

All in all, these results suggest that when direct PCA eigenvectors and neural network weights were unconstrained they converged to periodic square solutions. However, when constrained to be non-negative, the direct PCA, and the corresponding neural network weights, both converged to a hexagonal solution.

## Dependence of the result on the structure of the input

We investigated the effect of different inputs on the emergence of the grid structure in the networks' output. We found that some manipulation of the input was necessary in orderto enable the

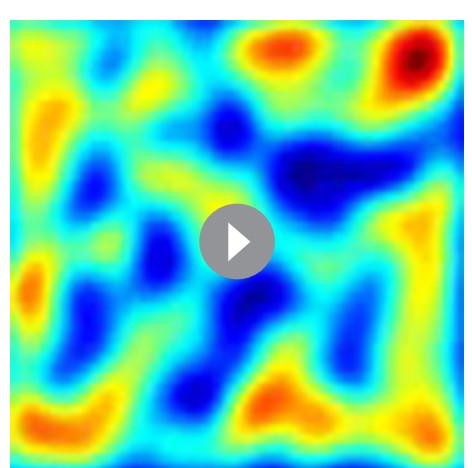

**Video 1.** Evolution in time of the network's weights. 625 Place-cells used as input. Video frame shown every 3000 time steps up to t=1,000,000. Video converges to results similar to those of *Figure 5*.

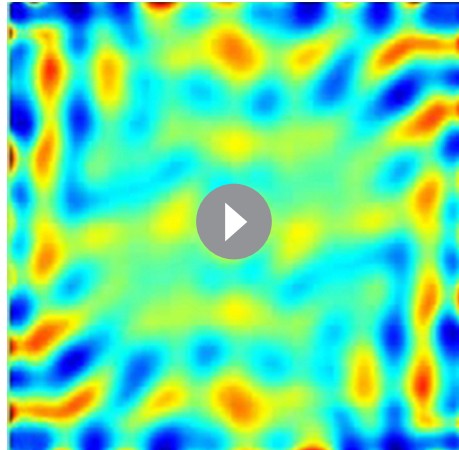

**Video 2.** Evolution of autocorrelation pattern of network's weights shown in *Video 1*.

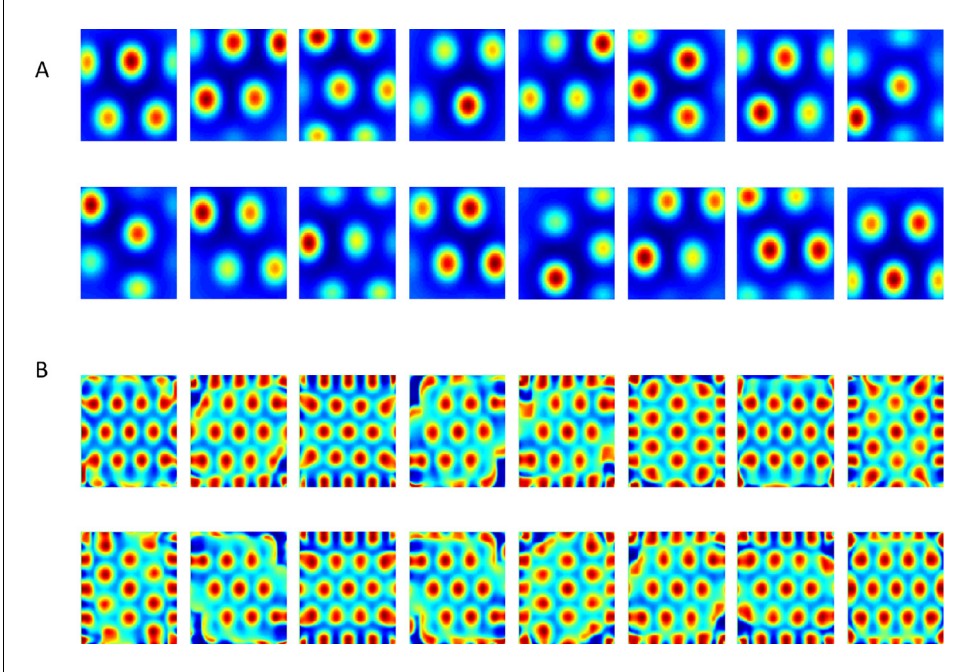

**Figure 6.** Results from the non-negative PCA algorithm. (**A**) Spatial projection of the leading eigenvector on input space. (**B**) Corresponding spatial autocorrelations. The different solutions are outcomes of multiple simulations with identical settings in a new environment and new random initial conditions.

implementation of PCA in the neural network. Specifically, PCA requires a zero-mean input, while simple Gaussian-like place cells do not possess this property. In order to obtain input with zero-mean, we either performed differentiation of the place cells' activity in time, or used a Mexican-hat like (Laplacian) shape (See Materials and methods for more details on the different types of inputs). Another option we explored was the usage of positive-negative disks with a total sum of zero activity in space (*Figure 8*). The motivation for the use of Mexican-hat like transformations is their abundance in the nervous system (*Wiesel and Hubel, 1963*; *Enroth-Cugell and Robson, 1966*; *Derdikman et al., 2003*).

We found that usage of simple 2-D Gaussian-functions as inputs did not generate hexagonal grid cells as outputs (*Figure 9*). On the other hand, time-differentiated inputs, positive-negative disks or Laplacian inputs did generate grid-like output cells, both when running the non-negative PCA directly (*Figure 6*), or by simulating the non-negatively constrained Neural Network (*Figure 5*). Another approach we used for obtaining zero-mean was to subtract the mean dynamically from every output individually (see Materials and methods). The latter approach, related to adaptation of the firing rate, was adopted from Kropff & Treves (*Kropff and Treves, 2008*), who used it to control various aspects of the grid cell's activity. In addition to controlling the firing rate of the grid cells, if applied correctly, the adaptation could be exploited to keep the output's activity stable, with zero-mean rates. We applied this method in our system and in this case the outputs converged to hexagonal grid cells as well, similarly to the previous cases (e.g. derivative in time, or Mexican hats as inputs; data not shown).

In summary, two conditions were required for the neural network to converge to spatial solutions resembling hexagonal grid cells: (1) non-negativity of the feedforward weights and (2) an effective zero-mean of the inputs (in time or space).

## Stability analysis
### Convergence to hexagons from various initial spatial conditions
In order to numerically test the stability of the hexagonal solution, we initialized the network in different ways, randomly, using linear stripes, squares, rhomboids (squares on hexagonal lattice) and

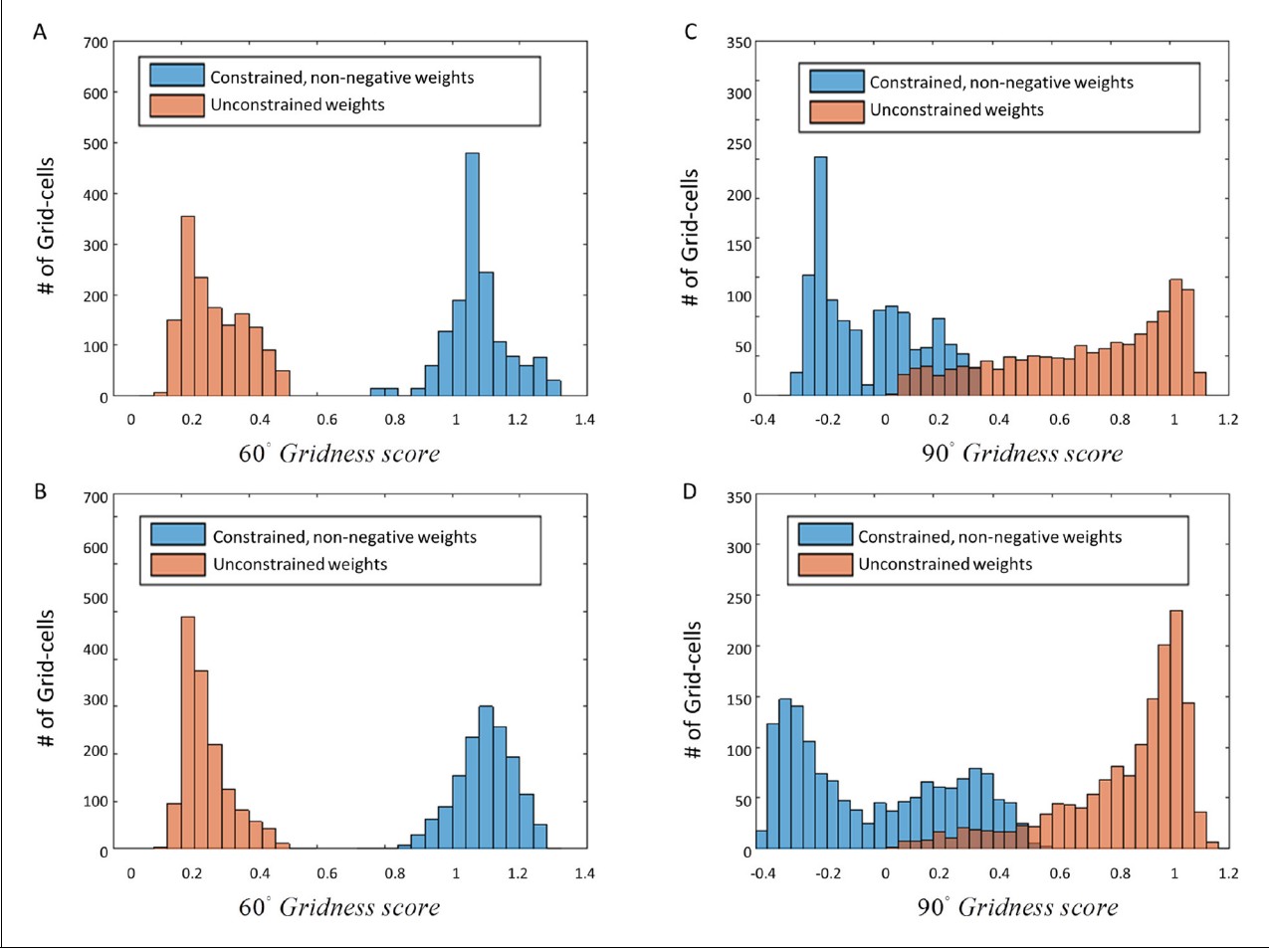

**Figure 7.** Histograms of Gridness values from network and PCA. First row (**A**) + (**C**) corresponds to network results, and second row (**B**) + (**D**) to PCA. The left column histograms contain the 60° Gridness scores and the right one the 90° Gridness scores.

noisy hexagons. In all cases, the network converged to a hexagonal pattern (*Figure 10*; for squares and stripes, other shapes not shown here).

We also ran the converged weights in a new simulation with novel trajectories and tested the Gridness scores, and the inter-trial stability in comparison to previous simulations. We found that the hexagonal solutions of the network remained stable although the trajectories varied drastically (data not shown).

## Asymptotic stability of the equilibria

Under certain conditions (e.g., decaying learning rates and independent and identically distributed (i.i.d.) inputs), it was previously proved (*Hornik and Kuan, 1992*), using techniques from the theory of stochastic approximation, that the system described here can be asymptotically analyzed in terms of (deterministic) Ordinary Differential Equations (ODE), rather than in terms of the stochastic recurrence equations. Since the ODE defining the converged weights is non-linear, we solved the ODEs numerically (see Materials and methods), by randomly initializing the weight vector. The asymptotic equilibria were reached much faster, compared to the outcome of the recurrence equations. Similarly to the recurrence equations, constraining the weights to be non-negative induced them to converge into a hexagonal shape while a non-constrained system produced square-like outcomes (*Figure 11*).

Simulation was run 60 times, with 400 outputs per run. 60° Gridness score mean was 1.1 ± 0.0006 when weights were constrained and 0.29 ± 0.0005 when weights were unconstrained. 90° Gridness

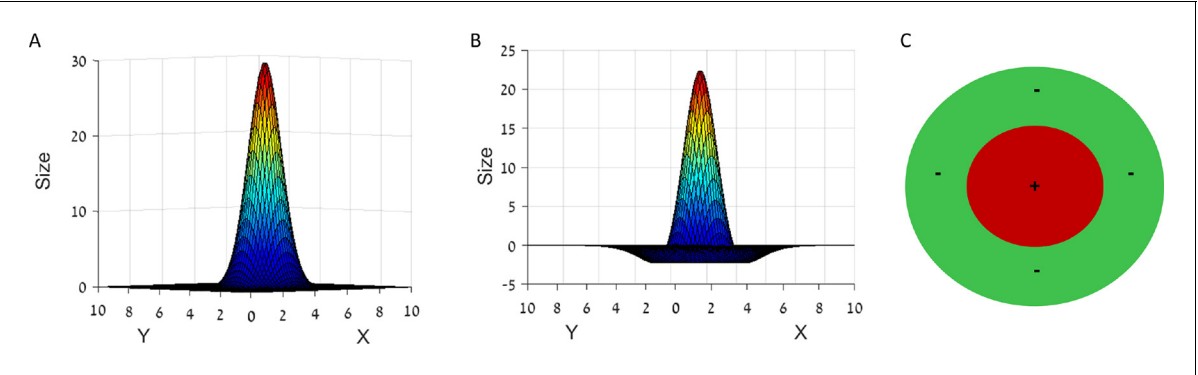

**Figure 8.** Different types of spatial input used in our network. (**A**) 2D Gaussian function, acting as a simple place cell. (**B**) Laplacian function or Mexican hat. (**C**) A positive (inner circle) - negative (outer ring) disk. While inputs as in panel A do not converge to hexagonal grids, inputs as in panels B or C do converge.

score mean was 0.006 ± 0.002 when weights were constrained and 0.8 ± 0.0017 when weights were unconstrained.

## Effect of place cell parameters on grid structure

A more detailed view of the resulting grid spacing showed that it was heavily dependent on the field widths of the place cells inputs. When the environment size was fixed and the output calculated per input size, the grid-spacing (distance between neighboring peaks) increased for larger place cell field widths.

To enable a fast parameter sweep over many place cell field widths (and large environment sizes), we took the steady state limit, and the limit of a high density of place cell locations, and used the fast FISTA algorithm to solve the non-negative PCA problem (see Materials and methods section).

We performed multiple simulations, and found that there was a simple linear dependency between the place field size and the output grid scale. For the case of periodic boundary conditions,

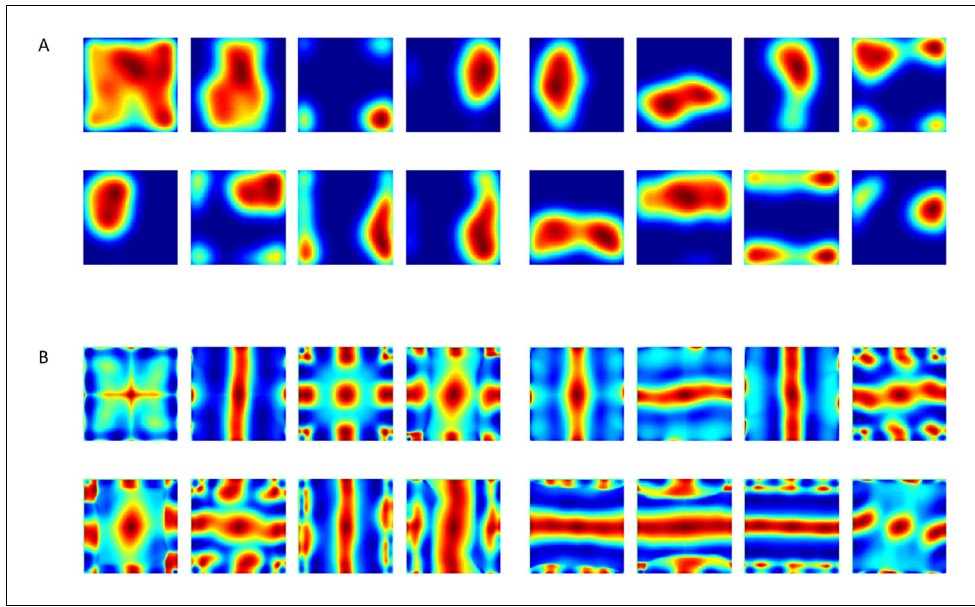

**Figure 9.** Spatial projection of outputs' weights in the neural network when inputs did not have zero mean (such as in *Figure 8A*). (**A**) Various weights plotted spatially as projection onto place cells space. (**B**) Autocorrelation of (**A**).

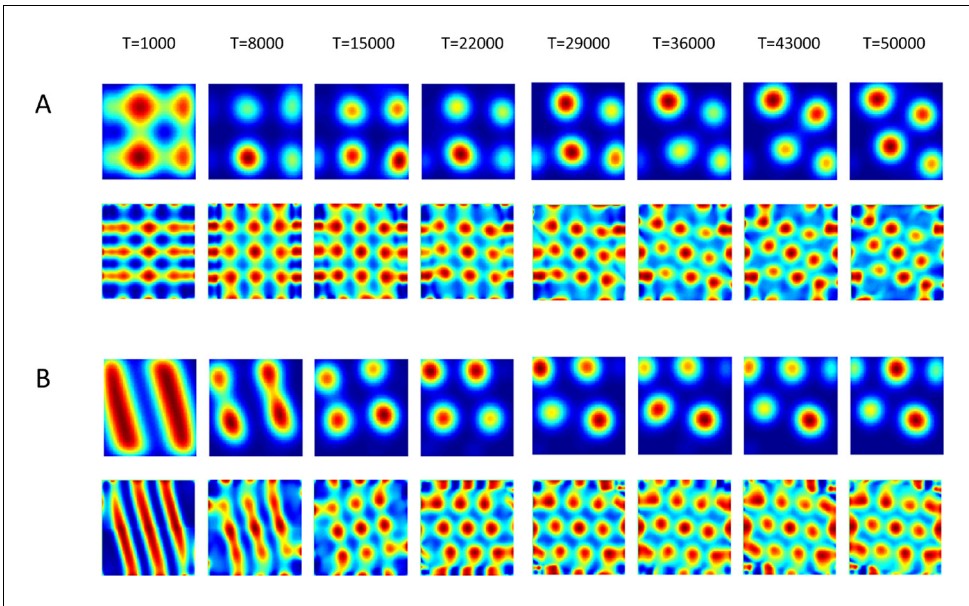

**Figure 10.** Evolution in time of the networks' solutions (upper rows) and their autocorrelations (lower rows). The network was initialized in shapes of (**A**) Squares and of (**B**) stripes (linear).

we found that grid scale was S = 7.5sigma+0.85, where sigma was the width of the place cell field (**Figure 12A**). For a different set of simulations with zero boundary conditions, we achieved a similar relation: S=7.54sigma+0.62 (figure not shown). Grid scale was more dependent on place field size and less on box size (**Figure 12H**). We note that for very large environments, the effects of boundary conditions diminishes. At this limit, this linear relation between place field size and grid scale can be explained from analytical considerations (see Materials and methods section). Intuitively, this follows from dimensional analysis: given an infinite environment, at steady state the length scale of the place cell field width is the only length scale in the model, so any other length scale must be proportional to this scale. More precisely, we can provide a lower bound for the linear fit (**Figure 12A**), which depends only on the tuning curve of the place cells (see Materials and methods section). This lower bound was derived for periodic boundary conditions, but works well even with zero boundary conditions (not shown).

Furthermore, we found that the grid orientation varied substantially for different place cell field widths, in the possible range of 0–15 degrees (**Figure 12C,D**). For small environments, the orientation strongly depended on the boundary conditions. However, as described in the Methods section, analytical considerations suggest that as the environment grows, the distribution of grid orientations becomes uniform in the range of 0–15 degrees, with a mean at 7.5°. Intuitively, this can be explained by rotational symmetry – when the environment size is infinite, all directions in the model are equivalent, and so we should get all orientations with equal probability, if we start the model from a uniformly random initialization. In addition, grid orientation was not a clear function of the gridness of the obtained grid cells (**Figure 12B**). For large enough place cells, gridness was larger than 1 (**Figure 12E–G**).

## Modules of grid cells

It is known that in reality grid cells form in modules of multiple spacings (**Barry et al., 2007**; **Stensola et al., 2012**). We tried to address this question of modules in several ways. First, we used different widths for the Gaussian/Laplacian input functions: Initially, we placed a heterogeneous population of widths in a given environment (i.e., uniformly random widths) and ran the single-output network 100 times. The distribution of grid spacings was almost comparable to the results of the largest width if applied alone, and did not exhibit module like behavior. This result is not surprising when thinking about a small place cell overlapping in space with a large place cell. Whenever the agent passes next to the small one, it activates both weights via synaptic learning. This causes the

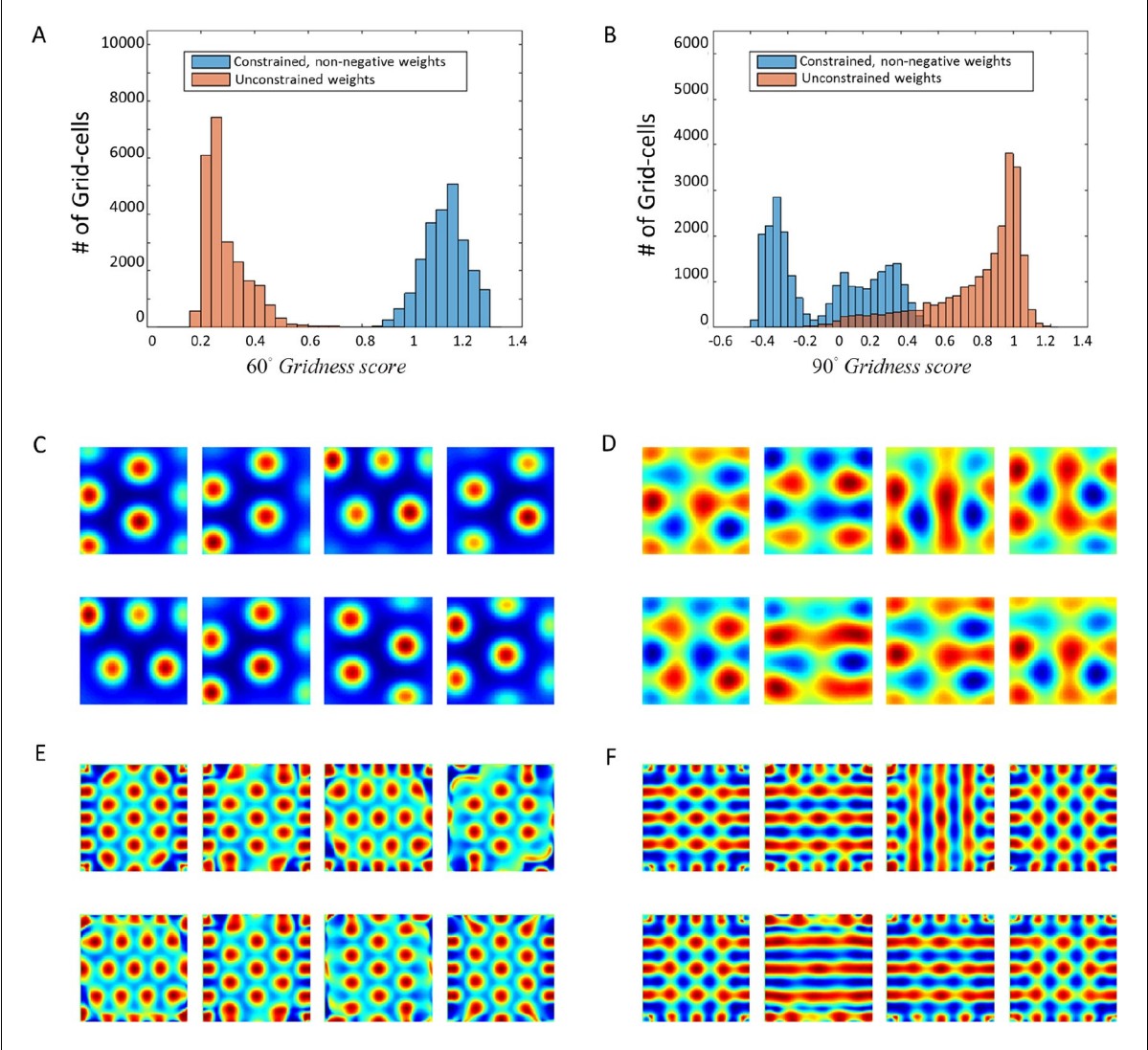

**Figure 11.** Numerical convergence of the ODE to hexagonal results when weights are constrained. (**A**) + (**B**): 60° and 90° Gridness score histograms. Each score represents a different weight vector of the solution J. (**C**) + (**D**): Spatial results for constrained and unconstrained scenarios, respectively. (**E**) + (**F**) Spatial autocorrelations of (**C**) + (**D**).

large firing field to overshadow the smaller one. Additionally, when using populations of only two widths of place fields, the grid spacings were dictated by the size of the larger place field (data not shown).

The second option we considered was to use a multi-output neural network, capable of computing all 'eigenvectors' rather than only the principal 'eigenvector' (where by 'eigenvector' we mean here the vectors achieved under the positivity constraint, and not the exact eigenvectors themselves). We used a hierarchical network implementation introduced by *Sanger, 1989* (see Materials and methods). Since the 1st output's weights converged to the 1st 'eigenvector', the network (*Figure 13A–B*) provided to the subsequent outputs (2nd, 3rd, and so forth) a reduced-version of the data from which the projection of the 1st 'eigenvector' has been subtracted out. This process, reminiscent of Gram-Schmidt orthogonalization, was capable of computing all 'eigenvectors' (in the modified sense) of the input's covariance matrix. It is important to note though that, due to the non-negativity constraint, the vectors achieved in this way were not orthogonal, and thus it cannot be considered a real orthogonalization process, although, as explained in the Methods section, the process does aim for maximum difference between the vectors.

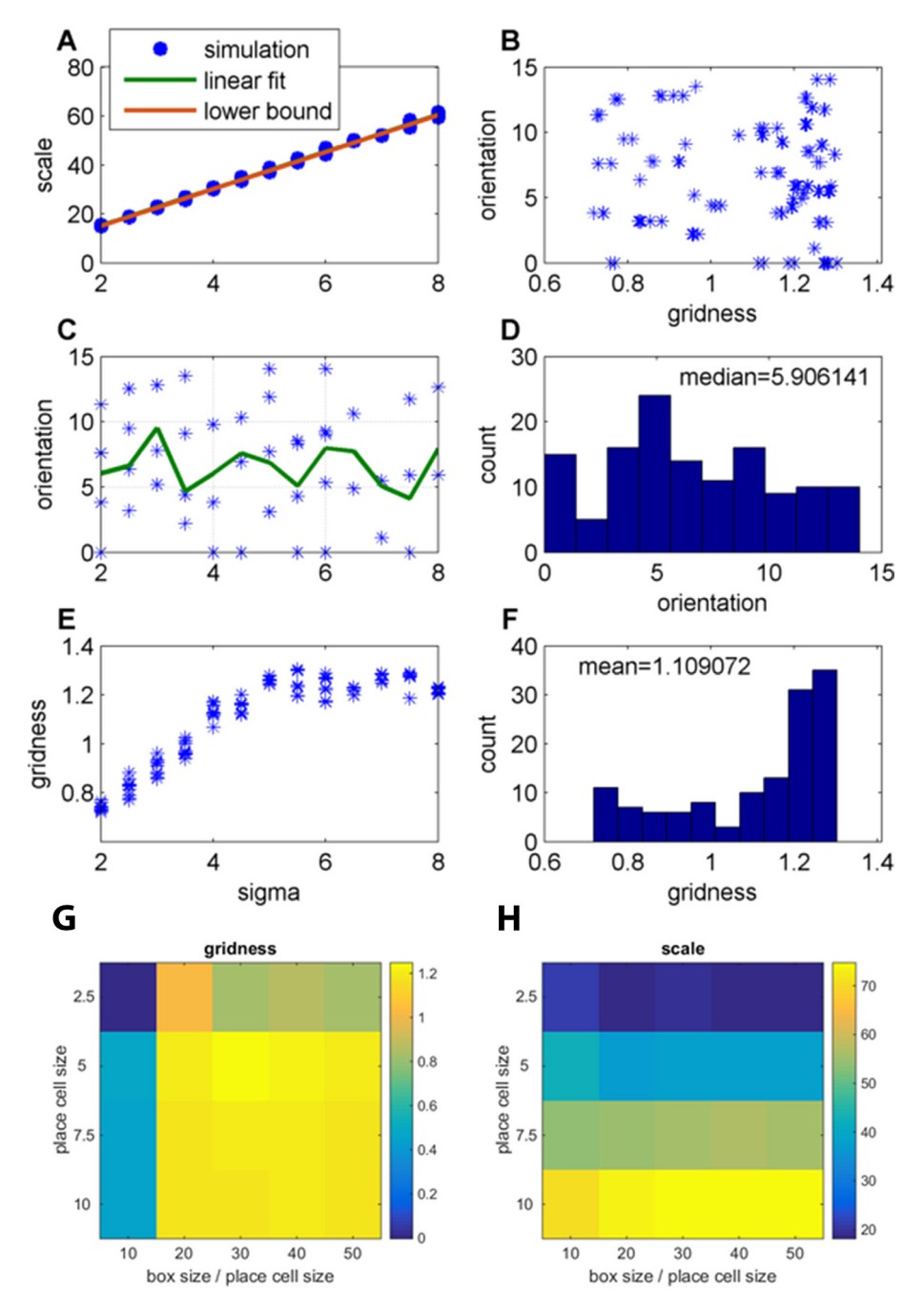

**Figure 12.** Effect of changing the place-field size in fixed arena (FISTA algorithm used; periodic boundary conditions and Arena size 500); (A) Grid scale as a function of place field size (sigma); Linear fit is: Scale = 7.4 Sigma+0.62; the lower bound, equal to $2\pi/k_{\dagger}$, were $k_{\dagger}$ is defined in **Equation 32** in the Materials and methods section; (B) Grid orientation as a function of gridness; (C) Grid orientation as a function of sigma – scatter plot (blue stars) and mean (green line); (D) Histogram of grid orientations; (E) Mean gridness as a function of sigma; and (F) Histogram of mean gridness. (G) Gridness as a function of sigma and (arena-size/sigma) (zero boundary conditions). (H) Grid scale for the same parameters as in G.

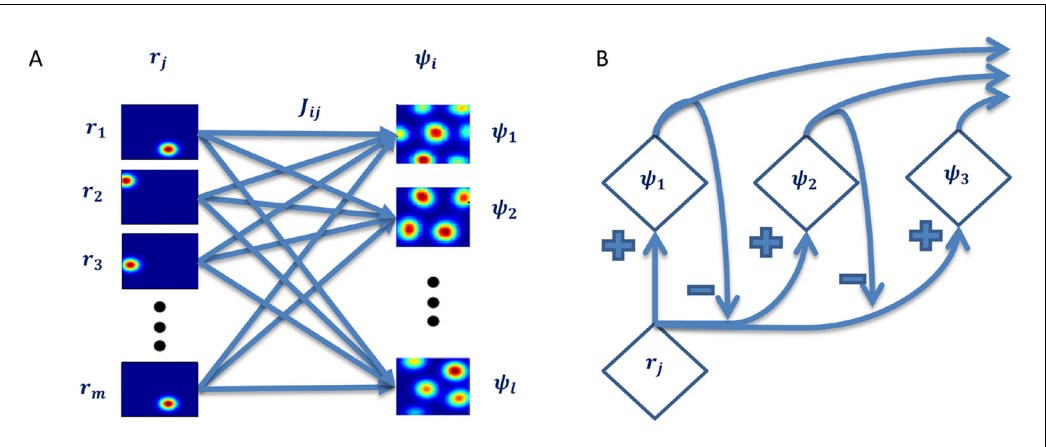

**Figure 13.** Hierarchial network capable of computing all 'principal components'. (A) Each output is a linear sum of all inputs weighted by the corresponding learned weights. (B) Over time, the data the following outputs 'see' is the original data after subtration of the 1st 'eigenvector's' projection onto it. This is an iterative process causing all outputs' weights to converge to the 'prinipcal components' of the data.

When constrained to be non-negative, and using the same homogeneous 'place cells' as in the previous network, the networks' weights converged to hexagonal shapes. Here, however, we found that the smaller the 'eigenvalue' was (or the higher the principal component number) the denser the grid became. We were able to identify two main populations of grid-distance 'modules' among the hexagonal spatial solutions with high Gridness scores (>0.7, *Figure 14A–B*). In addition, we found that the ratio between the distances of the modules was ~1.4, close to the value of 1.42 found by Stensola et al. (*Stensola et al., 2012*). Although we searched for additional such jumps, we could only identify this single jump, suggesting that our model can yield up to two 'modules' and not more. The same process was repeated using the direct PCA method, utilizing the covariance matrix of the data after simulation as input for the non-negative PCA algorithms, and considering their ability to calculate only the 1st 'eigenvector'. By iteratively projecting the 1st 'eigenvector' on the simulation data and subtracting the outcome from the original data, we applied the non-negative PCA algorithm to the residual data obtaining the 2nd 'eigenvector' of the original data. This 'eigenvector' now constituted the 1st eigenvector' of the new residual data (see Materials and methods). Applying this process to as many 'outputs' as needed, we obtained very similar results to the ones presented above using the neural network (data not shown).

## Discussion

In our work, we explored the nature and behavior of the feedback projections from place cells to grid cells. We shed light on the importance of this relation and showed, both analytically and in simulation, how a simple single-layer neural network could produce hexagonal grid cells when subjected to place cell-like temporal input from a randomly-roaming moving agent. We found that the network resembled a neural network performing PCA (*Oja, 1982*), with the constraint that the weights were non-negative. Under these conditions, and also under the requirements that place cells have a zero mean in time or space, the first principal component in the 2D arena had a firing pattern resembling a hexagonal grid cell. Furthermore, we found that in the limit of very large arenas, grid orientation converged to a uniform distribution in range of 0–15°. When looking at additional components, grid scale tends to be discretely clustered, such that two modules emerge. This is partially consistent with current experimental findings (*Stensola et al., 2012*; *2015*). Furthermore, the inhibitory connectivity between multiple grid cells is consistent with the known functional anatomy in this network (*Couey et al., 2013*).

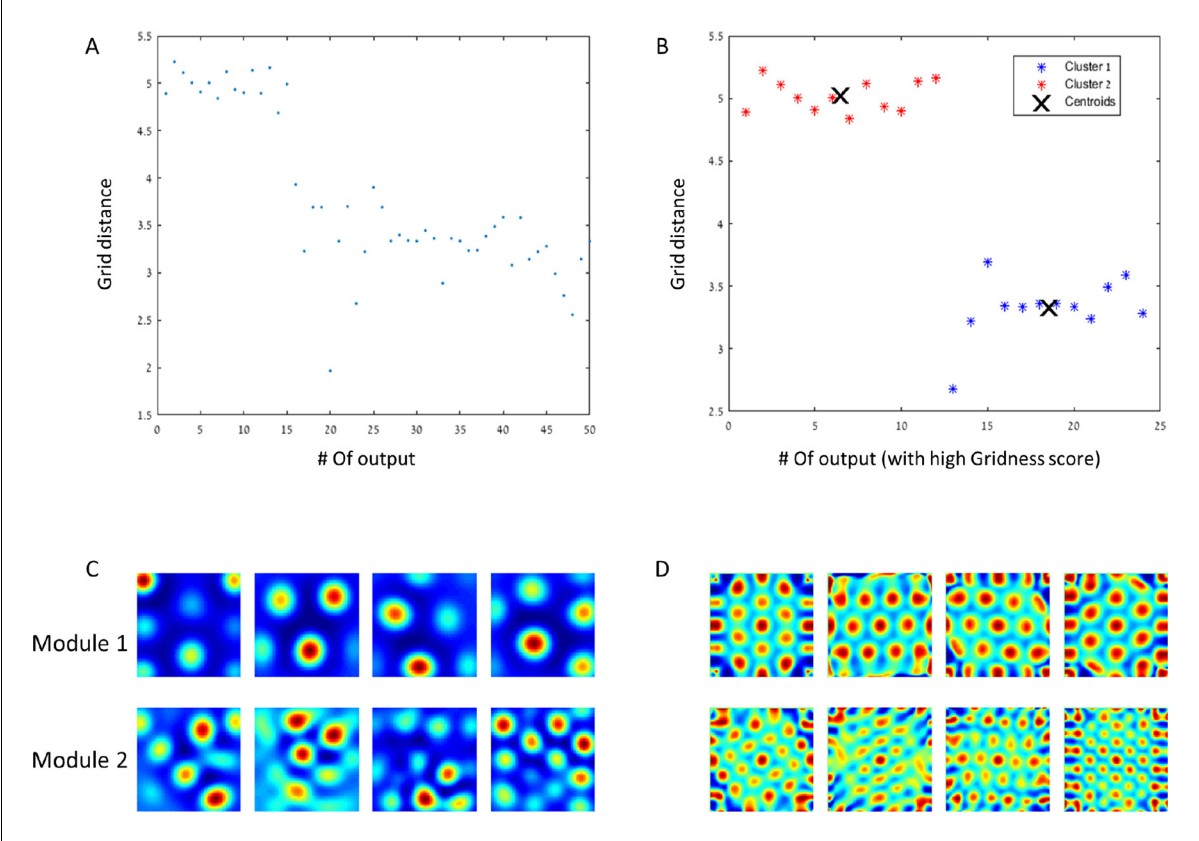

**Figure 14.** Modules of grid cells. (**A**) In a network with 50 outputs, the grid spacing per output is plotted with respect to the hierarchical place of the output. (**B**) The grid spacing of outputs with high Gridness score (>0.7). The centroids have a ratio of close to $\sqrt{2}$. (**C**) + (**D**) Example of rate maps of outputs and their spatial autocorrelations for both of the modules.

## Place-to-Grid as a PCA network

As a consequence of the requirements for PCA to hold, we found that the place cell input needed to have a zero-mean, otherwise the output was not periodic. Due to the lack of the zero-mean property in 2D Gaussians, we used various approaches to impose zero-mean on the input data. The first, in the time domain, was to differentiate the input and use the derivatives (a random walk produces zero-mean derivatives) as inputs. Another approach was to dynamically subtract the mean in all iterations of the simulation. This approach was reminiscent of the adaptation procedure suggested in the Kropff & Treves paper (*Kropff and Treves, 2008*). A third approach, applied in the spatial domain was to use inputs with a zero-spatial mean such as Laplacians of Gaussians (Mexican hats in 2D, or differences-of-Gaussians) or negative – positive disks. Such Mexican-hat inputs are quite typical in the nervous system (*Wiesel and Hubel, 1963*; *Enroth-Cugell and Robson, 1966*; *Derdikman et al., 2003*), although in the case of place cells it is not completely known how they are formed. They could be a result of interaction between place cells and the vast number of inhibitory interneurons in the local hippocampal network (*Freund and Buzsáki, 1996*).

Another condition we found crucial, which was not part of the original PCA network, was a non-negativity constraint on the place-to-grid learned weights. While rather easy to implement in the network, adding this constraint to the non-convex PCA problem was harder to implement. Since the problem is NP-hard (*Montanari and Richard, 2014*), we turned to numerical methods. We used three different algorithms (*Montanari and Richard, 2014*; *Zass and Shashua, 2006*; *Beck and Teboulle, 2009*) to find the leading 'eigenvector' of every given temporal based input. As shown in the results section, both processes (i.e. direct PCA and the neural network) resulted in hexagonal outcomes when the non-negativity and zero-mean criteria were met. Note that the ease of use of

the neural network for solving the positive PCA problem is a nice feature of the neural network implementation, and should be investigated further.

We also note that while our network focused on the projection from place cells to grid cells, we cannot preclude the importance of the reciprocal projection from grid cells to place cells. Further study will be needed to 'close the loop' and simultaneously consider both of these projections at once.

## Similar studies

We note that similar work has noticed the relation between place-cell-to-grid-cell transformation and PCA. Notably, *Stachenfeld et al., (2014)* have demonstrated, from considerations related to reinforcement learning, that grid cells could be related to place cells through a PCA transformation. However, due to the unconstrained nature of their transformation, the resulting grid cells were square-like. Furthermore, there has been an endeavor to model the transformation from place cells to grid cells using independent-component-analysis (*Franzius et al., 2007*).

We also note that there is now a surge of interest in the feedback projection from place cells to grid-cells, which is inverse to the anatomical downstream direction from grid cells to place cells (*Witter and Amaral, 2004*) that has guided most of the models to-date (*Zilli, 2012*; *Giocomo et al., 2011*). In addition to several papers from the Treves group, in which the projection from place cells to grid cells is studied (*Kropff and Treves, 2008*; *Si and Treves, 2013*), there has been also recent work from other groups as well exploring this direction (*Castro and Aguiar, 2014*; *Stepanyuk, 2015*). As far as we are aware, none of the previous studies noted the importance of the non-negativity constraint and the requirement of zero mean input. Additionally, to the best of our knowledge, the analytic results and insights provided in this work (see Materials and methods) are novel, and provide a mathematically consistent explanation for the emergence of hexagonally-spaced grid cells.

## Predictions of our model

Based on the findings of this work, it is possible to make several predictions. First, the grid cells must receive zero-mean input over time to produce hexagonally shaped firing patterns. With all feedback projections from place cells being excitatory, the lateral inhibition from other neighboring grid cells might be the balancing parameter to achieve the temporal zero-mean (*Couey et al., 2013*). Alternatively, an adaptation method, such as the one suggested in *Kropff and Treves, (2008)* may be applied. Second, if indeed the grid cells are a lower dimensional representation of the place cells in a PCA form, the place-to-grid neural weights distribution should be similar across identically spaced grid cell populations. This is because all grid cells with similar spacing would have maximized the variance over the same input, resulting in similar spatial solutions. As an aside, we note that such a projection may be a source of phase-related correlations in grid cells (*Tocker et al., 2015*). Third, we found a linear relation between the size of the place cells and the spacing between grid cells. Furthermore, the spacing of the grid cells is mostly determined by the size of the largest place cell – predicting that the feedback from large place cells is not connected to grid cells with small spacing. Fourth, we found modules of different grid spacings in a hierarchical network with the ratio of distances between successive units close to $\sqrt{2}$. This result is in accordance with the ratio reported in *Stensola et al., (2012)*. However, we note that there is a difference between our results and experimental results because the analysis predicts that there should only be two modules, while the data show at least 5 modules, with a range of scales, the smallest and most numerous having approximately the scale of the smaller place fields found in the dorsal hippocampus (25–30 cm). Fifth, for large enough environments our model suggests that, from mathematical considerations, the grid orientation should approach a uniform orientation in the possible range of 0–15°. This is in discrepancy with experimental results which measure a peak at 7.5°, and not a uniform distribution (*Stensola et al., 2015*). As noted, the discrepancies between our results and reality may relate to the fact that a more advanced model will have to take into account both the downstream projection from grid cells to place cells together with the upstream projection from place cells to grid cells discussed in this paper. Furthermore, such a model will have to take into account the non-uniform distribution of place-cell widths (*Kjelstrup et al., 2008*).

## Why hexagons?

In light of our results, we further asked what is special about the hexagonal shape which renders it a stable solution. Past works have demonstrated that hexagonality is optimal in terms of efficient coding. Two recent papers have addressed the potential benefit of encoding by grid cells. *Mathis et al., (2015)* considered the decoding of spatial information based on a grid-like periodic representation. Using lower bounds on the reconstruction error based on a Fisher information criterion, they demonstrated that hexagonal grids lead to the highest spatial resolution in two dimensions (extensions to higher dimensions were also provided). The solution is obtained by mapping the problem onto a circle packing problem. The work of *Wei et al., (2013)* also took a decoding perspective, and showed that hexagonal grids minimize the number of neurons required to encode location with a given resolution. Both papers offer insights into the possible information theoretic benefits of the hexagonal grid solution. In the present paper, we were mainly concerned with a specific biologically motivated learning (development) mechanism that may yield such a solution. Our analysis suggests that the hexagonal patterns can arise as a solution that maximizes the grid cell output variance, under non-negativity constraints. In Fourier space, the solution is a hexagonal lattice with lattice constant near the peak of the Fourier transform of the place cell tuning curve (*Figures 15* and *16*; see Materials and methods).

To conclude, this work demonstrates how grid cells could be formed from a simple Hebbian neural network with place cells as inputs, without needing to rely on path-integration mechanisms.

## Materials and methods

All code was written in MATLAB, and can be obtained on https://github.com/derdikman/Dordek-et-al.-Matlab-code.git or on request from authors.

### Neural network architecture

We implemented a single-layer neural network with feedforward connections that was capable of producing a hexagonal-like output (*Figure 2*). The feedforward connections were updated according to a self-normalizing version of a Hebbian learning rule referred to as the Oja rule (*Oja, 1982*),

$$\Delta J_i^t = \varepsilon^t(\psi^t r_i^t - (\psi^t)^2 J_i^t), \tag{1}$$

where $\varepsilon^t$ denotes the learning rate, $J_i^t$ is the $i^{th}$ weight and $\psi^t, r_i^t$ are the output and the $i^{th}$ input of the network, respectively (all at time $t$). The weights were initialized randomly according to a uniform distribution and then normalized to have norm 1. The output $\psi^t$ was calculated every iteration by summing up all pre-synaptic activity from the entire input neuron population. The activity of each output was processed through a sigmoidal function (e.g., $\tanh$) or a simple linear function. Formally,

$$\psi^t = f\left(\sum_{i=1}^n J_i^t \cdot r_i^t\right), \tag{2}$$

where n is the number of input place cells. Since we were initially only concerned with the eigenvector associated with the largest eigenvalue, we did not implement a multiple-output architecture. In this formulation, in which no lateral weights were used, multiple outputs were equivalent to running the same setting with one output several times.

As discussed in the introduction, this kind of simple feedforward neural network with linear activation and a local weight update in the form of Oja's rule (1) is known to perform *Principal Components Analysis (PCA)* (*Oja, 1982*; *Sanger, 1989*; *Weingessel and Hornik, 2000*). In the case of a single output the feedforward weights converge to the principal eigenvector of the input's covariance matrix. With several outputs, and lateral weights, as described in the section on modules, the weights converge to the leading principal eigenvectors of the covariance matrix, or, in certain cases (*Weingessel and Hornik, 2000*), to the subspace spanned by the principal eigenvectors. We can thus compare the results of the neural network to those of the mathematical procedure of PCA. Hence, in our simulation, we (1) let the neural networks' weights develop in real time based on the current place cell inputs. In addition, we (2) saved the input activity for every time step to calculate the input covariance matrix and perform (batch) PCA directly.

It is worth mentioning that the PCA solution described in this section can be interpreted differently based on the Singular Value Decomposition (SVD). Denoting by $R$ the $T \times d$ spatio-temporal

pattern of place cell activities (after setting the mean to zero), where $T$ is the time duration and $d$ is the number of place cells, the SVD decomposition (see *Jolliffe, 2002*; sec. 3.5) for $R$ is $R = \mathrm{ULA'}$. For a matrix $R$ of rank $r$, $L$ is a $r \times r$ diagonal matrix whose $k$th element is equal to $l_k^{1/2}$, the square root of the $k$th eigenvalue of the covariance matrix $\mathrm{RR'}$ (computed in the PCA analysis), $A$ is the $d \times r$ matrix with $k$th column equal to the $k$th eigenvector of $\mathrm{RR'}$, and $U$ is the $T \times r$ matrix whose $k$th column is $l_k^{-1/2} R a_k$. Note that $U$ is a $T \times r$ dimensional matrix whose $k$th column represents the temporal dynamics of the $k^{\text{th}}$ grid cell. In other words, the SVD provides a decomposition of the place cell activity in terms of the grid cell activity, as opposed to the grid cell representation in terms of place cell activity we discussed so far. The network learns the spatial weights over place cells (the eigenvectors) as the connections weights from the place cells, and 'projection onto place cell space' ($l_k^{-1/2} R a_k$) is simply the firing rates of the output neuron plotted against the location of the agent.

The question we therefore asked was under what conditions, when using *place cell-like* inputs, a solution resembling hexagonal *grid cells* emerges. To answer this we used both the neural-network implementation and the direct calculation of the PCA coefficients.

## Simulation

We simulated an agent moving in a 2D virtual environment consisting of a square arena covered by $n$ uniformly distributed 2D Gaussian-shaped place cells, organized on a grid, given by

$$r_i^t\left(\boldsymbol{X}(t)\right) = \exp\left(\frac{-\left(\boldsymbol{X}(t) - \boldsymbol{C}_i\right)^2}{2\sigma_i^2}\right), \qquad i = 1, 2, ..., n \tag{3}$$

where $\boldsymbol{X}(t)$ represents the location of the agent. The variables $r_i^t$ constitute the temporal input from place cell $i$ at time $t$, and $\boldsymbol{C}_i, \sigma_i$ are the $i^{th}$ place cell's field center and width, respectively (see variations on this input structure below). In order to eliminate boundary effects, periodic boundary conditions were assumed. The virtual agent moved about in a random walk scheme (see Appendix) and explored the environment (*Figure 1A*). The place cell centers were assumed to be uniformly distributed (*Figure 1B*) and shared the same standard deviation $\sigma$. The activity of all place cells as a function of time $(r(t)_1, r(t)_2 \ldots r(t)_n)$ was dependent on the stochastic movement of the agent, and formed a [*Neuron x Time*] matrix ($r \in R^{nxT}$, with T- being the time dimension, see *Figure 1C*).

The simulation was run several times with different input arguments (see *Table 1*). The agent was simulated for $T$ time steps, allowing the neural network's weights to develop and reach a steady state by using the learning rule (*Equations 1,2*) and the input (*Equation 3*) data. The simulation parameters are listed below and include parameters related to the environment, simulation, agent and network variables.

To calculate the PCA directly, we used the MATLAB function **Princomp** in order to evaluate the $n$ principal eigenvectors $\{\vec{q}_k\}_{k=1}^n$ and corresponding eigenvalues of the input covariance matrix. As mentioned in the Results section, there exists a near fourfold redundancy in the eigenvectors (X-Y axis and in phase). *Figure 3* demonstrates this redundancy by plotting the eigenvalues of the covariance matrix. The output response of each eigenvector $\vec{q}_k$ corresponding to a 2D input location $(x, y)$ is

$$\Phi(x,y)_k = \sum_{j=1}^n q_k^j \exp\left(-\frac{(x - c_x^j)^2}{2\sigma_x^2} - \frac{(y - c_y^j)^2}{2\sigma_y^2}\right), \qquad k = 1, 2, ..., \ n \tag{4}$$

**Table 1.** List of variables used in simulation.

| Environment: | Size of arena | Place cells field width | Place cells distribution |
|---|---|---|---|
| Agent: | Velocity (angular & linear) | Initial position | ———————— |
| Network: | # Place cells/ #Grid cells | Learning rate | Adaptation variable (if used) |
| Simulation: | Duration (time) | Time step | ———————— |

where $c_x^j$ and $c_y^j$ are the $x, y$ components of the centers of the individual place cell fields. Unless otherwise mentioned, we used place cells in a rectangular grid, such that a place cell is centered at each pixel of the image (that is – number of place cells equals the number of image pixels).

## Non-negativity constraint

Projections between place cells and grid cells are known to be primarily excitatory (*Witter and Amaral, 2004*), thus if we aim to mimic the biological circuit, a non-negativity constraint should be added to the feedforward weights in the neural network. While implementing a non-negativity constraint in the neural network is rather easy (a simple rectification rule in the weight dynamics, such that weights which are smaller than 0 are set to 0), the equivalent condition for calculating non-negative Principal Components is more intricate. Since this problem is non-convex and, in general, NP-hard (*Montanari and Richard, 2014*), a numerical procedure was imperative. We used three different algorithms for this purpose.

The first (*Zass and Shashua, 2006*) named NSPCA (Nonnegative Sparse PCA) is based on coordinate-descent. The algorithm computes a non-negative version of the covariance matrix's eigenvectors and relies on solving a numerical optimization problem, converging to a local maximum starting from a random initial point. The local nature of the algorithm did not guarantee a convergence to a global optimum (recall that the problem is non-convex). The algorithm's inputs consisted of the place cell activities' covariance matrix, $\alpha$ - a balancing parameter between reconstruction and orthonormality, $\beta$ – a variable which controls the amount of sparseness required, and an initial solution vector. For the sake of generality, we set the initial vector to be uniformly random (and normalized), $\alpha$ was set to a relatively high value – $10^4$ and since no sparseness was needed, $\beta$ was set to zero.

The second algorithm (*Montanari and Richard, 2014*) does not require any simulation parameters except an arbitrary initialization. It works directly on the inputs and uses a message passing algorithm to define an iterative algorithm to approximately solve the optimization problem. Under specific assumptions it can be shown that the algorithm asymptotically solves the problem (for large input dimensions).

The third algorithm we use is the parameter free Fast Iterative Threshold and Shrinkage algorithm FISTA (*Beck and Teboulle, 2009*). As described later in this section, this algorithm is the fastest of the three, and allowed us rapid screening of parameter space.

## Different variants of input structure

Performing PCA on raw data requires the subtraction of the data mean. Some thought was required in order to determine how to perform this subtraction in the case of the neural network.

One way to perform the subtraction in the time domain was to dynamically subtract the mean during simulation by using the discrete 1st or 2nd derivatives of the inputs in time [i.e. from *Equation 3*, $\Delta r(t+1) = r(t+1) - r(t)$]. Under conditions of an isotropic random walk (namely, given any starting position, motion in all directions is equally likely) it is clear that $E[\Delta r(t)] = 0$. Another option for subtracting the mean in the time domain was the use of an adaptation variable, as was initially introduced by *Kropff and Treves, (2008)*. Although originally exploited for control over the firing rate, it can be viewed as a variable that represents subtraction of a weighted sum of the firing rate history. Instead of using the inputs $r_i^t$ directly in *Equation 2* to compute the activation $\psi^t$, an intermediate adaptation variable $\psi_{adp}^t(\delta)$ was used ($\delta$ being the relative significance of the present temporal sample) as

$$\psi_{adp}^t = \psi^t - \overline{\psi}^t, \tag{5}$$

$$\overline{\psi}^t = (1-\delta) \cdot \overline{\psi}^{t-1} + \delta \psi^t. \tag{6}$$

It is not hard to see that for *i.i.d.* variables $\psi_{adp}^t$, the sequence $\overline{\psi}^t$ converges for large $t$ to the mean of $\psi^t$. Thus, when $t \longrightarrow \infty$ we find that $E[\psi_{adp}^t] \longrightarrow 0$, specifically, the adaptation variable is of zero asymptotic mean.

The second method we used to enforce a zero mean input was simply to create it in advance. Rather than using 2D Gaussian functions (i.e. [*Equation 3*]) as inputs we used 2D difference-of-Gaussians (all $\sigma$ are equal in x and y axis):

$$r_i^t\left(\boldsymbol{X}(t)\right) = c_{1,i} \exp\left(-\frac{\left(\boldsymbol{X}(t) - \boldsymbol{C}_i\right)^2}{2\sigma_{1,i}^2}\right) - c_{2,i} \exp\left(-\frac{\left(\boldsymbol{X}(t) - \boldsymbol{C}_i\right)^2}{2\sigma_{2,i}^2}\right), \qquad i = 1, 2, \dots, n \tag{7}$$

where the constants $c_1$ and $c_2$ are set so the integral of the given Laplacian function is zero (if the environment size is not too small, then $c_{1,i}/c_{2,i} \approx \sigma_{2,i}/\sigma_{1,i}$). Therefore, if we assume a random walk that covers the entire environment uniformly, the temporal mean of the input would be zero as well. Such input data can be inspired by similar behavior of neurons in the retina and the lateral-geniculate nucleus (*Wiesel and Hubel, 1963*; *Enroth-Cugell and Robson, 1966*). Finally, we implemented another input data type; positive-negative disks (see Appendix). Analogously to the difference-of-Gaussians function, the integral over input is zero so the same goal (zero-mean) was achieved. It is worthwhile noting that subtracting a constant from a simple Gaussian function is not sufficient since at infinity it does not reach zero.

## Quality of solution and Gridness

In order to test the hexagonality of the results we used a hexagonal *Gridness score* (*Sargolini et al., 2006*). The Gridness score of the spatial fields was calculated from a cropped ring of their autocorrelogram including the six maxima closest to the center. The ring was rotated six times, $30°$ per rotation, reaching in total angles of $30°, 60°, 90°, 120°, 150°$. Furthermore, for every rotated angle the Pearson correlation with the original un-rotated map was obtained. Denoting by $C_\gamma$ the correlation for a specific rotation angle $\gamma$, the final Gridness score was (*Kropff and Treves, 2008*):

$$Gridness \quad = \quad \frac{1}{2}(C_{60} + C_{120}) - \frac{1}{3}(C_{30} + C_{90} + C_{150}). \tag{8}$$

In addition to this 'traditional' score we used a *Squareness* Gridness score in order to examine how square-like the results are spatially. The special reference to the square shape was driven by the tendency of the spatial solution to converge to a rectangular shape when no constrains were applied. The Squareness Gridness score is similar to the hexagonal one, but now the cropped ring of the autocorrelogram is rotated $45°$ every iteration to reach angles of $45°, 90°, 135°$. As before, denoting $C_\gamma$ as the correlation for a specific rotation angle $\gamma$ the new Gridness score was calculated as:

$$Square\ Gridness = C_{90} - \frac{1}{2}(C_{45} + C_{135}). \tag{9}$$

All errors calculated in gridness measures are SEM (Standard Error of the Mean).

## Hierarchical networks and modules

As described in the Results section, we were interested to check whether a hierarchy of outputs could explain the module phenomenon described for real grid cells. We replaced the single-output network with a hierarchical, multiple outputs network, which is capable of computing all 'principal components' of the input data while maintaining the non-negativity constraint as before. The network, introduced by *Sanger, 1989*, computes each output as a linear summation of the weighted inputs similar to *Equation 2*. However, the weights are now calculated according to:

$$\Delta J_{ij}^t = \varepsilon^t(r_j^t \psi_i^t - \psi_i^t \sum_{k=1}^{i} J_{kj}^t \psi_k^t). \tag{10}$$

The first term in the parenthesis when $k = 1$ was the regular Hebb-Oja derived rule. In other words, the first output calculated the first non-negative 'principal component' (in inverted commas due to the non-negativity) of the data. Following the first one, the weights of each output received a back projection from the previous outputs. This learning rule applied to the data in a similar manner to the Gram-Schmidt process, subtracting the 'influence' of the previous 'principal components' on the data and recalculating the appropriate 'principal components' of the updated input data.

In a comparable manner, we applied this technique to the input data $\boldsymbol{X}$ in order to obtain non-negative 'eigenvectors' from the direct nonnegative-PCA algorithms. We found $\boldsymbol{V}_2$ by subtracting from the data the projection of $\boldsymbol{V}_1$ on it,

$$\tilde{X} = X - V_1^T (V_1 \cdot X). \tag{11}$$

Next, we computed $V_2$, the first non-negative 'principal component' of $\tilde{X}$, and similarly the subsequent ones.

## Stability of hexagonal solutions

In order to test the stability of the solutions we obtained under all types of conditions, we applied the ODE method (*Kushner and Clark, 1978*; *Hornik and Kuan, 1992*; *Weingessel and Hornik, 2000*) to the PCA feature extraction algorithm introduced in pervious sections. This method allows one to asymptotically replace the stochastic update equations describing the neural dynamics by smooth differential equations describing the average asymptotic behavior. Under appropriate conditions, the stochastic dynamics converge with probability one to the solution of the ODEs. Although originally this approach was designed for a more general architecture (including lateral connections and asymmetric updating rules), we used a restricted version for our system. In addition, the following analysis is accurate solely for linear output functions. However, since our architecture works well with either linear or non-linear output functions, the conclusions are valid.

We can rewrite the relevant updating equations of the linear neural network (in matrix form), (see [*Weingessel and Hornik, 2000*] *Equations 15–19*):

$$\psi^{t+1} = Q \cdot J^t \cdot (r^t)^T, \tag{12}$$

$$\Delta J^t = \varepsilon^t \left( \psi^t (r^t)^T - \Phi(\psi^t \cdot (\psi^t)^T) J^t \right). \tag{13}$$

In our case we set

$$Q = I, \qquad \Phi = diag.$$

Consider the following assumptions

1. The input sequence $r^t$ consists of independent identically distributed, bounded random variables with zero-mean.
2. $\{\varepsilon^t\}$ is a positive number sequence satisfying: $\sum_t \varepsilon^t = \infty, \sum_t (\varepsilon^t)^2 < \infty$.

A typical suitable sequence is $\varepsilon^t = \frac{1}{t}, t = 1, 2 \dots$.

For long times, we denote

$$E[\psi^t (r^t)^T] \longrightarrow E[J \cdot r \cdot r^T] = E[J] \cdot E[r \cdot r^T] = J\Sigma, \tag{14}$$

$$\lim_{t \to \infty} E[\psi \psi^T] = E[J] \cdot E[rr^T] \cdot E[J^T] = J\Sigma J^T. \tag{15}$$

The penultimate equalities in these equations used the fact that the weights converge with probability one to their average value, resulting from the solution of the ODEs. Following *Weingessel and Hornik, (2000)*, we can analyze *Equations 12,13* under the above assumptions, via their asymptotically equivalent associated ODEs

$$\frac{dJ}{dt} = J\Sigma - diag(J\Sigma J^T) J, \tag{16}$$

with equilibria at

$$J\Sigma = diag(J\Sigma J^T) J. \tag{17}$$

We solved it numerically by exploiting the same covariance matrix and initializing with random weights $J$. In line with our previous findings, we found that constraining $J$ to be non-negative (by a simple cut-off rule) resulted in a hexagonal shape (in the projection of $J$ onto the place cells space; *Figure 11*). In contrast, when the weights were not constrained they converged to square-like results.

## Steady state analysis

From this point onwards, we focus on the case of a single output, in which $J$ is a row vector, unless stated otherwise. In the unconstrained case, from *Equation 17* any $J$ which is a normalized

eigenvector of $\Sigma$ would be a fixed point. However, from *Equation 16*, only the principal eigenvector, which is the solution to the following optimization problem

$$\max_{J: J^T J = 1} J\Sigma J^T \tag{18}$$

would correspond to a stable fixed point. This is the standard PCA problem. By adding the constraint $J \geq 0$ we get the non-negative PCA problem.

To speed up simulation and simplify analysis we make further simplifications.

First, we assume that the agent's random movement is ergodic (e.g., an isotropic random walk in a finite box as we used in our simulation), uniform and covering the entire environment, so that

$$J\Sigma J^T = E\left[\psi^2\left(X(t)\right)\right] = \frac{1}{|S|}\int_S \psi^2(\mathbf{x})d\mathbf{x}, \tag{19}$$

where $\mathbf{x}$ denotes location vector (in contrast to $X(t)$, which is the random process corresponding to the location of the agent), $S$ is the entire environment, and $|S|$ is the size of the environment.

Second, we assume that the environment $S$ is uniformly and densely covered by identical place cells, each of which has the same a tuning curve $r(\mathbf{x})$ (which integrates to zero). In this case, the activity of the linear grid cell becomes a convolution operation

$$\psi(\mathbf{x}) = \int_S J(\mathbf{x}')r(\mathbf{x}-\mathbf{x}')d\mathbf{x}', \tag{20}$$

where $J(\mathbf{x})$ is the synaptic weight connecting to the place cell at location $\mathbf{x}$.

Thus, we can write our objective as

$$\frac{1}{|S|}\int_S \psi^2(\mathbf{x})d\mathbf{x} = \frac{1}{|S|}\int_S \left(\int_S J(\mathbf{x}')r(\mathbf{x}-\mathbf{x}')d\mathbf{x}'\right)^2 d\mathbf{x} \tag{21}$$

under the constraint that the weights are normalized

$$\frac{1}{|S|}\int_S J^2(\mathbf{x})\,d\mathbf{x} = 1, \tag{22}$$

where either $J(\mathbf{x}) \in \mathbb{R}$ (PCA) or $J(\mathbf{x}) \geq 0$ ('non-negative PCA').

Since we expressed the objective using a convolution operation (different boundary conditions can be assumed), it can be solved numerically considerably faster. In the non-negative case, we used the parameter free Fast Iterative Threshold and Shrinkage algorithm [FISTA (*Beck and Teboulle, 2009*); in which we do not use shrinkage, since we only have hard constraints], where the gradient was calculated efficiently using convolutions.

Moreover, as we show in the following sections, if we assume periodic boundary conditions and use Fourier analysis, we can analytically find the PCA solutions, and obtain important insight on the non-negative PCA solutions.

## Fourier notation

Any continuously differentiable function $f(\mathbf{x})$, defined over $S \triangleq [0, L]^D$, a 'box' region in $D$ dimensions, with periodic boundary conditions, can be written using a Fourier series

$$\hat{f}(\mathbf{k}) \triangleq \frac{1}{|S|}\int_S f(\mathbf{x})e^{i\mathbf{k}\cdot\mathbf{x}}d\mathbf{x},$$

$$f(\mathbf{x}) \triangleq \sum_{\mathbf{k}\in\hat{S}}\hat{f}(\mathbf{k})e^{-i\mathbf{k}\cdot\mathbf{x}}, \tag{23}$$

where $|S| = L^D$ is the volume of the box and

$$\hat{S} \triangleq \left\{\left(\frac{2m_1\pi}{L}, \ldots, \frac{2m_d\pi}{L}\right)\right\}_{(m_1,\ldots,m_d)\in\mathbb{Z}^D}$$

is the reciprocal lattice of $S$ in $\mathbf{k}$-space (frequency space).

## PCA solution

Assuming periodic boundary conditions, we use Parseval's identity, and the properties of the convolution, to transform the steady state objective (*Equation 21*) to its simpler form in the Fourier domain,

$$\frac{1}{|S|}\int_S \psi^2(\mathbf{x})\mathrm{d}\mathbf{x} = \sum_{\mathbf{k}\in\hat{S}} |\hat{J}(\mathbf{k})\hat{r}(\mathbf{k})|^2. \tag{24}$$

Similarly, the normalization constraint can also be written in the Fourier domain,

$$\frac{1}{|S|}\int_S J^2(\mathbf{x})\mathrm{d}\mathbf{x} = \sum_{\mathbf{k}\in\hat{S}} |\hat{J}(\mathbf{k})|^2 = 1. \tag{25}$$

Maximizing the objective *Equation 24* under this constraint in the Fourier domain, we immediately get that any solution is a linear combination of the Fourier components,

$$\hat{J}(\mathbf{k}) = \begin{cases} 1, & \mathbf{k}=\mathbf{k}_* \\ 0, & \mathbf{k}\neq\mathbf{k}_* \end{cases}. \tag{26}$$

where

$$\mathbf{k}_* \in \mathrm{argmax}_{\mathbf{k}\in\hat{S}}\,\hat{r}(\mathbf{k}), \tag{27}$$

and $\hat{J}(\mathbf{k})$ satisfies the normalization constraint. In the original space, the Fourier components are

$$J(\mathbf{x}) = e^{i\mathbf{k}\cdot\mathbf{x}+i\phi}, \tag{28}$$

where $\phi \in [0, 2\pi)$ is a free parameter that determines the phase. Also, since $J(\mathbf{x})$ should assume real values, it is composed of real Fourier components

$$J(\mathbf{x}) = \frac{1}{\sqrt{2}}(e^{i\mathbf{k}_*\cdot\mathbf{x}+i\phi} + e^{-i\mathbf{k}_*\cdot\mathbf{x}-i\phi}) = \sqrt{2}\cos(\mathbf{k}_*\cdot\mathbf{x}+\phi). \tag{29}$$

This is a valid solution, since $r(\mathbf{x})$ is a real-valued function, $\hat{r}(\mathbf{k}) = \hat{r}(-\mathbf{k})$ and therefore $-\mathbf{k}_* \in \mathrm{argmax}_{\mathbf{k}\in\hat{S}}\hat{r}(\mathbf{k})$.

## PCA solution for a difference of Gaussians tuning curve

In this paper we focused on the case where $r(\mathbf{x})$ has the shape of a difference of Gaussians (*Equation 7*),

$$r(\mathbf{x}) \propto c_1 \exp\left(-\frac{\|\mathbf{x}\|^2}{2\sigma_1^2}\right) - c_2 \exp\left(-\frac{\|\mathbf{x}\|^2}{2\sigma_2^2}\right) \tag{30}$$

where $c_1$ and $c_2$ are some positive normalization constants, set so that $\int_S r(\mathbf{x})d\mathbf{x} = 0$ (see appendix). The Fourier transform of $r(\mathbf{x})$ is also a difference of Gaussians

$$\hat{r}(\mathbf{k}) \propto \exp\left(-\frac{1}{2}\sigma_1^2\|\mathbf{k}\|^2\right) - \exp\left(-\frac{1}{2}\sigma_2^2\|\mathbf{k}\|^2\right) \tag{31}$$

$\forall\mathbf{k} \in \hat{S}$, as we show in the appendix. Therefore the value of the Fourier domain objective only depends on the radius $\|\mathbf{k}\|$, and all solutions $\mathbf{k}_*$ have the same radius $\|\mathbf{k}_*\|$. If $L\longrightarrow\infty$, then the $k$-lattice $\hat{S}$ becomes dense ($\hat{S}\longrightarrow\mathbb{R}^D$) and this radius is equal to

$$\mathbf{k}_\dagger = \mathrm{argmax}_{\mathbf{k}\geq 0}\left[\exp\left(-\frac{1}{2}\sigma_1^2 k^2\right) - \exp\left(-\frac{1}{2}\sigma_2^2 k^2\right)\right] \tag{32}$$

which is a unique maximizer, that can be easily obtained numerically.

Notice that if we multiply the place cell field width by some positive constant $c$, then the solution $k_\dagger$ will be divided by $c$. The grid spacing, proportional to $\frac{1}{k_\dagger}$, would therefore also be multiplied by $c$.

This entails a linear dependency between the place cell field width and the grid cell spacing, in the limit of a large box size ($L\longrightarrow\infty$). When the box has a finite size, $k$-lattice discretization also has a (usually small) effect on the grid spacing.

In that case, all solutions $\mathbf{k}_*$ are restricted to be on the finite lattice $\hat{S}$. Therefore, the solutions $\mathbf{k}_*$ are the points on the lattice $\hat{S}$ for which the radius $\|\mathbf{k}_*\|$ is closest to $k_\dagger$ (see **Figure 15B,C**).

## The degeneracy of the PCA solution

The number of real-valued PCA solutions (degeneracy) in 1D is two, as there are exactly two maxima, $k_*$ and $-k_*$. The phase $\phi$, determines how the components at $k_*$ and $-k_*$ are linearly combined.

However, there are more maxima in the 2D case. Specifically, given a maximum $\mathbf{k}_*$, we can write $(m, n) = \frac{L}{2\pi}\mathbf{k}_*$, where $(m, n) \in \mathbb{Z}^2$. Usually there are 7 other different points with the same radius: $(m, -n), (-m, -n), (-m, n), (-n, -m), (n, -m), (-n, m)$ and $(n, m)$, so we will have a degeneracy of eight (corresponding to the symmetries of a square box). This is case of points in group B, shown in **Figure 15C**.

However, we can also get a different degeneracy. First, if either $m = \pm n$, $n = 0$ or $m = 0$ we will have a degeneracy of 4, since then some of the original eight points will coincide (groups A,C and D in **Figure 15C**). Second, additional points $(k, r)$ can exist such that $k^2 + r^2 = m^2 + n^2$, (Pythagorean triplets with the same hypotenuse) – for example, $15^2 + 20^2 = 25^2 = 7^2 + 24^2$. These points will also appear in groups of four or eight.

Therefore, we will always have a degeneracy which is some multiple of 4. Note that in the full network simulation, the degeneracy is not exact. This is due to the perturbation noise from the agent's random walk as well as the non-uniform sampling of the place cells.

## The PCA solution with a non-negative constraint

Next, we add the non-negativity constraint $J(\mathbf{x}) \geq 0$. As mentioned earlier, this constraint renders the optimization problem NP-hard, and prevents us from a complete analytical solution. We therefore combine numerical and mathematical analysis, in order to gain intuition as to why

1. Locally optimal 2D solutions are hexagonal.
2. These solutions have a grid spacing near $(4\pi/(\sqrt{3}k_\dagger)$ ($k_\dagger$ is the peak of $\hat{r}(k)$).

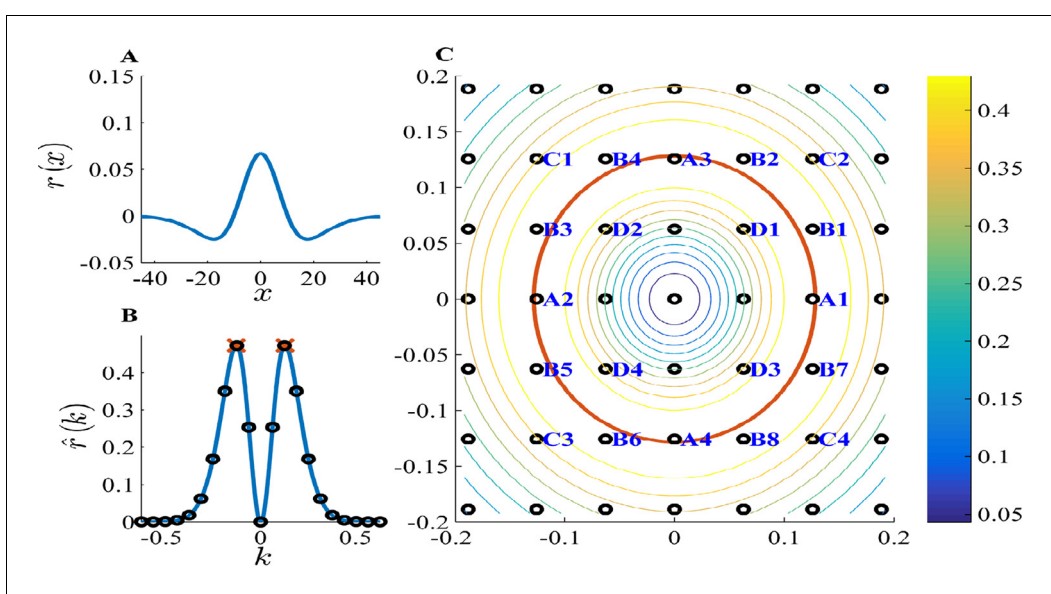

**Figure 15.** PCA k-space analysis for a difference of Gaussians tuning curve. (A) The 1D tuning curve $r(x)$. (B) The 1D tuning curve Fourier transform $\hat{r}(k)$. The black circles indicate k-lattice points. The PCA solution, $k_*$, is given by the circles closest to $k_\dagger$, the peak of $\hat{r}(k)$ (red cross). (C) A contour plot of the 2D tuning curve Fourier transform $\hat{r}(\mathbf{k})$. In 2D k-space the peak of $\hat{r}(\mathbf{k})$ becomes a circle (red), and the k-lattice $\hat{S}$ is a square lattice (black circles). The lattice point can be partitioned into equivalent groups. Several such groups are marked in blue on the lattice. For example, the PCA solution Fourier components lie on the four lattice points closest to the circle, denoted A1-4. Note the grouping of A,B,C & D (4,8,4 and 4, respectively) corresponds to the grouping of the 20 highest principal components in **Figure 4**. Parameters: $2\sigma_1 = \sigma_2 = 7.5, L = 100$.

3. The average grid alignment is approximately 7.5°, for large environments.
4. Why grid cells have modules, and what is their spacing.

## 1D Solutions

Our numerical results indicate that the Fourier components of any locally optimal 1D solution of non-negative PCA have the following structure:

1. There is a non-negative 'DC component' ($k = 0$).
2. The maximal non-DC component, ($k \neq 0$) is $k_*$, where $k_*$ is 'close' (more details below) to $k_\dagger$, the peak of $\hat{r}(k)$.
3. All other non-zero Fourier components are $\{mk_*\}_{n=1}^{\infty}$, weaker harmonies of $k_*$.

This structure suggests that the component at $k_*$ aims to maximize the objective, while the other components guarantee the non-negativity of the solution $J(x)$. In order to gain some analytical intuition as to why this is the case, we first examine the limit case that $L \longrightarrow \infty$ and $\hat{r}(k)$ is highly peaked at $k_\dagger$. In that case the Fourier objective (**Equation 24**) simply becomes $2|\hat{r}(k_\dagger)|^2|\hat{J}(k_\dagger)|^2$. For simplicity, we will rescale our units so that $|\hat{r}(k_\dagger)|^2 = 1/2$, and the objective becomes $|\hat{J}(k_\dagger)|^2$. Therefore, the solution must include a Fourier component at $k_\dagger$ or the objective would be zero. The other components exist only to maintain the non-negativity constraint, since if they increase in magnitude, then the objective, which is proportional to $|\hat{J}(k_\dagger)|^2$, must decrease to compensate (due to the normalization constraint – **Equation 25**). Note that these components must include a positive 'DC component' at $k = 0$, or else $\int_S J(x)dx \propto \hat{J}(0) \leq 0$, which contradicts the constraints. To find all the Fourier components, we examine a solution composed of only a few ($M$) components

$$J(x) = \hat{J}(0) + 2\sum_{m=1}^{M} \hat{J}_m \cos(k_m x + \phi_m).$$

Clearly, we can set $k_1 = k_\dagger$, or otherwise, the objective would be zero. Also, we must have

$$\hat{J}(0) = -\min_x \left(2\sum_{m=1}^{M} \hat{J}_m \cos(k_m x + \phi_m)\right) \geq 0.$$

Otherwise, the solution would be either (1) negative or (2) non-optimal, since we can decrease $\hat{J}(0)$ and increase $|J_1|$.

For $M = 1$, we immediately get that, in the optimal solution, $2\hat{J}_1 = \hat{J}(0) = \sqrt{2/3}$ ($\phi_m$ does not matter). For $M = 2, 3$ and $4$ a solution is harder to find directly, so we performed a parameter grid search over all the free parameters ($k_m, \hat{J}_m$ and $\phi_m$) in those components. We found that the optimal solution (which maximizes the objective $|\hat{J}(k_\dagger)|^2$), had the following form

$$J(x) = \sum_{m=-M}^{M} \hat{J}(mk_\dagger) \cos\left(mk_\dagger(x - x_0)\right), \tag{33}$$

where $x_0$ is a free parameter. This form results from a parameter grid search for $M = 1, 2, 3$ and $4$, under the assumption that $L \longrightarrow \infty$ and $\hat{r}(k)$ is highly peaked. However, our numerical results in the general case (**Figure 16A**), using the FISTA algorithm, indicate that the locally optimal solution does not change much even if $L$ is finite, and $\hat{r}(k)$ is not highly peaked. Specifically, it has a similar form

$$J(x) = \sum_{m=-\infty}^{\infty} \hat{J}(mk_*) \cos\left(mk_*(x - x_0)\right). \tag{34}$$

Since $\hat{J}(mk_*)$ is rapidly decaying (**Figure 16A**), effectively only the first few components are non-negligible, as in **Equation 33**. This can also be seen in the value of the objective obtained in the parameter scan

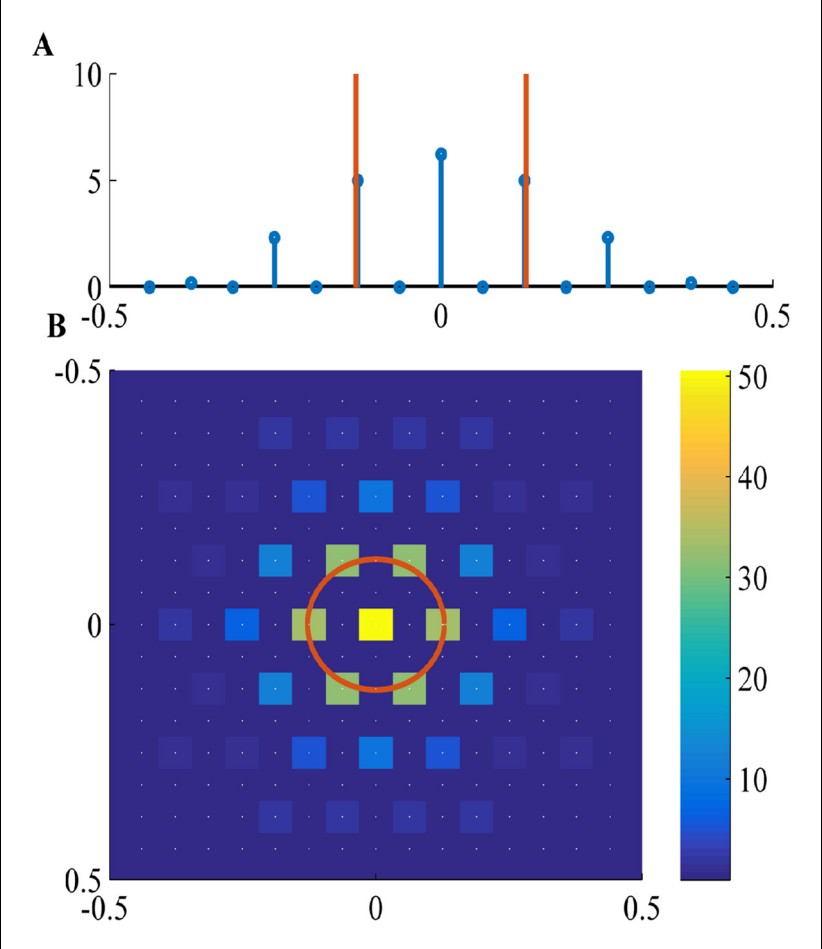

**Figure 16.** Fourier components of Non-negative PCA on the $k$-lattice. (**A**) 1D solution (blue) includes: a DC component ($k = 0$), a maximal component with magnitude near $k_{\dagger}$ (red line), and weaker harmonics of the maximal component. (**B**) 2D solution includes: a DC component ($\boldsymbol{k} = (0,0)$), a hexgaon of strong components with radius near $k_{\dagger}$ (red circle), and weaker components on the lattice of the strong components. White dots show underlying $k$-lattice. We used a difference of Gaussians tuning curve, with parameters $2\sigma_1 = \sigma_2 = 7.5, L = 100$, and the FISTA algorithm.

$$
\begin{array}{cccccc}
M & 1 & 2 & 3 & 4 \\
|\hat{J}(k_{\dagger})|^2 & \frac{1}{6} & 0.2367 & 0.2457 & 0.2457 \,,
\end{array}
\tag{35}
$$

where the contribution of additional high frequency components to the objective quickly becomes negligible. In fact, the value of the objective cannot increase above $0.25$, as we explain in the next section.

And so, the main difference between *Equations 33 and 34* is the base frequency, $k_*$, which is slightly different from $k_{\dagger}$. As explained in the appendix, the relation between $k_*$ and $k_{\dagger}$ depends on the $k$-lattice discretization, as well as on the properties of $\hat{r}(k)$.

## 2D Solutions

The 1D properties, described in the previous section, generalize to the 2D case in the following manner:

1. There is a non-negative DC component ($\mathbf{k} = (0,0)$).

2. A small 'basis set' of components $\left\{\mathbf{k}_*^{(i)}\right\}_{i=1}^{B}$ with similar amplitudes, and with similar radii $\left\|\mathbf{k}_*^{(i)}\right\|$ which are all 'close' to $k_\dagger$ (details below).

3. All other non-zero Fourier components are weaker, and restricted to the lattice

$$\left\{\mathbf{k} \in \hat{S} \mid \mathbf{k} = \sum_{i=1}^{B} n_i \mathbf{k}_*^{(i)}\right\}_{(n_1,\dots,n_B) \in \mathbb{Z}^B}$$

Interestingly, given these properties of the solution we already get hexagonal patterns, as we explain next.

Similarly to the 1D case, the difference between $\left\|\mathbf{k}_*^{(i)}\right\|$ and $k_\dagger$ is affected by lattice discretization, and the curvature of $\hat{r}(k)$ near $k_\dagger$. To simplify matters, we focus first on the simple case that $L \longrightarrow \infty$ and $\hat{r}(k)$ is sharply peaked around $k_\dagger$. Therefore, the Fourier objective becomes $\sum_{i=1}^{B} |\hat{J}\left(\mathbf{k}_*^{(i)}\right)|^2$, so the only Fourier components that appear in the objective are $\{\mathbf{k}_*^{(i)}\}_{i=1}^{B}$, which have radius $k_\dagger$. We examine the values this objective can have.

All the base components have the same radius. This implies, according to the Crystallographic restriction theorem in 2D, that the only allowed lattice angles (in the range between 0 and 90 degrees) are 0, 60 and 90 degrees. Therefore, there are only three possible lattice types in 2D. Next, we examine the value of the objective for each of these lattice types:

1) Square lattice, in which $\mathbf{k}_*^{(1)} = k_\dagger(1,0)$, $\mathbf{k}_*^{(2)} = k_\dagger(0,1)$, up to a rotation. In this case,

$$J(x,y) = \sum_{m_x=-\infty}^{\infty} \sum_{m_y=-\infty}^{\infty} \hat{J}_{m_x,m_y} \cos\left(k_\dagger(m_x x + m_y y) + \phi_{m_x,m_y}\right)$$

and the value of the objective is bounded above by $0.25$ (see proof in appendix).

2) 1D lattice, in which $\mathbf{k}_*^{(1)} = k_\dagger(1,0)$, up to a rotation. This is a special case of the square lattice, with a subset of $\hat{J}_{m_x,m_y}$ equal to zero, so we can write, as we did in the 1D case

$$J(x) = \sum_{m=-\infty}^{\infty} \hat{J}_m \cos(k_\dagger m x + \phi_m)$$

Therefore, the same objective upper bound, $0.25$, holds. Note that some of the solutions we found numerically are close to this bound (*Equation 35*).

3) Hexagonal lattice, in which the base components are

$$\mathbf{k}_*^{(1)} = k_\dagger(1,0), \mathbf{k}_*^{(2)} = k_\dagger\left(-\frac{1}{2}, \frac{\sqrt{3}}{2}\right), \mathbf{k}_*^{(3)} = k_\dagger\left(-\frac{1}{2}, -\frac{\sqrt{3}}{2}\right)$$

up to a rotation by some angle $\alpha$. Our parameter scans indicate that the objective value cannot surpass $0.2$ in any solution composed of only the base hexgonal components $\{\mathbf{k}_*^{(m)}\}_{m=1}^{3}$ and a DC component. However, taking into account also some higher order lattice components, we can find a better solution, with an objective value of $0.2558$. Though this is not necessarily the optimal solution, it surpasses any possible solutions on the other lattice types (bounded below $0.25$, as we proved in the appendix). Specifically, this solution is composed of the base vectors $\{\mathbf{k}_*^{(m)}\}_{m=1}^{3}$ and their harmonics

$$J(\mathbf{x}) = \hat{J}_0 + 2 \sum_{m=1}^{8} \hat{J}_m \cos(\mathbf{k}_*^{(m)} \cdot \mathbf{x})$$

with $\mathbf{k}_*^{(4)} = 2\mathbf{k}_*^{(1)}$, $\mathbf{k}_*^{(5)} = 2\mathbf{k}_*^{(2)}$, $\mathbf{k}_*^{(6)} = 2\mathbf{k}_*^{(3)}$, $\mathbf{k}_*^{(7)} = \mathbf{k}_*^{(1)} + \mathbf{k}_*^{(2)}$, $\mathbf{k}_*^{(8)} = \mathbf{k}_*^{(1)} + \mathbf{k}_*^{(3)}$. Also, $\hat{J}_0 = 0.6449$, $\hat{J}_1 = \hat{J}_2 = \hat{J}_3 = 0.292$, $\hat{J}_4 = \hat{J}_5 = \hat{J}_6 = -0.0101$ and $\hat{J}_7 = \hat{J}_8 = -0.134$.

Thus, any optimal solution must be on the hexagonal lattice, given our approximations. In practice, the lattice hexagonal basis vectors do not have exactly the same radius, and, as in the 1D case, this radius is somewhat smaller then $k_\dagger$, due to the lattice discretization, and due to that $\hat{r}(\mathbf{k})$ is not sharply peaked. However, the resulting solution lattice is still approximately hexagonal in $k$-space.

For example, this can be seen in the numerically obtained solution in *Figure 16B* – where the strongest non-DC Fourier components form an approximate hexagon near $k_\dagger$, from the Fourier components A, defined in *Figure 17*.

## Grid spacing

In general, we get a hexagonal grid pattern in $x$-space. If all base Fourier components have a radius of $k_\dagger$, then the grid spacing in $x$-space would be $4\pi/(\sqrt{3}k_\dagger)$. Since the radius of the basis vectors can be smaller than $k_\dagger$, the value of $4\pi/(\sqrt{3}k_\dagger)$ is a lower bound to the actual grid spacing (as demonstrated in *Figure 12A*), up to lattice discretization effects.

## Grid alignment

The angle of the hexagonal grid, $\alpha$, is determined by the directions of the hexagonal vectors. An angle $\alpha$ is possible, if there exists a $k$-lattice point $\mathbf{k} = \frac{2\pi}{L}(m, n)$ (with $m, n$ integers), for which $\frac{\pi}{L} > \sqrt{\frac{2\pi}{L}(m^2 + n^2) - k_*^2}$, and then $\alpha = \arctan\frac{n}{m}$. Since the hexagonal lattice has rotational symmetry of $60°$, we can restrict $\alpha$ to be in the range $-30° \leq \alpha \leq 30°$. The grid alignment, which is the minimal angle of the grid with the box boundaries is given by

$$\text{Grid alignment} = \min\Big(|\alpha|, 90° - (|\alpha| + 60°)\Big) = \min(|\alpha|, 30° - |\alpha|) \tag{36}$$

which is limited to the range $[0°, 15°]$, since $-30° \leq \alpha \leq 30°$. There are usually several possible grid alignments which are (approximately) rotated versions of each other (i.e., different $\alpha$). Note that, due to the $k$-lattice discretization, different alignments can result in slightly different objective values. However, the numerical algorithms we used to solve the optimization problem reached many possible grid alignments with a positive probability (*Figure 12C*), since we started from a random initialization and converged to a local minimum.

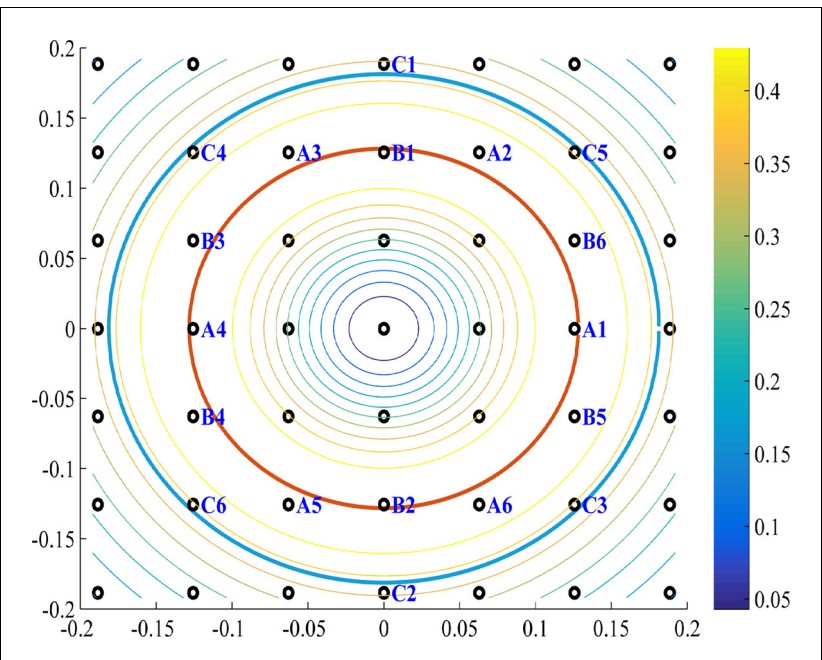

**Figure 17.** The modules in Fourier space. As in *Figure 15C*, we see a contour plot of the 2D tuning curve Fourier transform $\hat{r}(\mathbf{k})$ and the k-space the peak of $\hat{r}(\mathbf{k})$ (red circle), and the k-lattice $\hat{S}$ (black circles). The lattice points can be divided into approximately hexagonal shaped groups. Several such groups are marked in blue on the lattice. For example, group A and B are optimal since they are nearest to the red circle. The next best (with the highest-valued contours) group of points, which have an approximate hexgonal shape, is C. Note that group C has a k-radius of approximately the optimal radius times $\sqrt{2}$ (cyan circle). Parameters: $2\sigma_1 = \sigma_2 = 7.5, L = 100$.

In the limit $L \longrightarrow \infty$, the grid alignment will become uniform in the range $[0°, 15°]$, and the average grid alignment is $7.5°$.

## Hierarchical networks and modules

There are multiple routes to generalize non-negative PCA with multiple vectors. In this paper we chose to do so using a 'Gramm-Schmidt' like process, which can be written in the following way. First we define,

$$r_1(\mathbf{x},\mathbf{y}) = r(\mathbf{x}-\mathbf{y}) = c_1 \exp\left(-\frac{\|\mathbf{x}-\mathbf{y}\|^2}{2\sigma_1^2}\right) - c_2 \exp\left(-\frac{\|\mathbf{x}-\mathbf{y}\|^2}{2\sigma_2^2}\right),$$

$$J_0(\mathbf{x}) = 0,$$

(37)

and then, recursively, this process recovers non-negative 'eigenvectors' by subtracting out the previous components, similarly to Sanger's multiple PCA algorithm (*Sanger, 1989*), and enforcing the non-negativity constraint.

$$r_{n+1}(\mathbf{x},\mathbf{y}) = r_n(\mathbf{x},\mathbf{y}) - \int_s r_n(\mathbf{x},\mathbf{z}) J_n(\mathbf{z}) J_n(\mathbf{y}) d\mathbf{z}$$

$$J_n(\mathbf{y}) = \arg\max_{J(\mathbf{x})\geq 0, \frac{1}{|S|}\int_s J^2(\mathbf{x})d\mathbf{x}=1} \int_s d\mathbf{x} \left(\int_s r_n(\mathbf{x},\mathbf{y}) J(\mathbf{y}) d\mathbf{y}\right)^2.$$

(38)

To analyze this, we write the objectives we maximize in the Fourier domain, using Parseval's Theorem.

For $n = 1$, we recover the old objective (*Equation 24*):

$$\sum_{\mathbf{k}\in\hat{S}} \left|\hat{r}(\mathbf{k})\hat{J}(\mathbf{k})\right|^2.$$

(39)

For $n = 2$, we get

$$\sum_{\mathbf{k}\in\hat{S}} |\hat{r}(\mathbf{k})\left(\hat{J}(\mathbf{k}) - \hat{J}_1(\mathbf{k})\left(\sum_{\mathbf{q}\in\hat{S}} \hat{J}_1^*(\mathbf{q})\hat{J}(\mathbf{q})\right)\right)|^2,$$

(40)

where $\hat{J}^*$ is the complex conjugate of $\hat{J}$. This objective is similar to the original one, except that it penalizes $\hat{J}(\mathbf{k})$ if its components are similar to those of $\hat{J}_1(\mathbf{k})$. As $n$ increases the objective becomes more and more complicated, but as before, it contains terms which penalize $\hat{J}_n(\mathbf{k})$ if its components are similar to any of the previous solutions (*i.e.*, $\hat{J}_m(\mathbf{k})$ for $m < n$). This form suggests that each new 'eigenvector' tends to occupy new points in the Fourier lattice (similarly to unconstrained PCA solutions).

For example, the numerical solution shown in *Figure 16B* is composed of the Fourier lattice components in group A, defined in *Figure 17*. A completely equivalent solution would be in group B (it is just a 90 degrees rotation of the first). The next 'eigenvectors' should then include other Fourier-lattice components outside groups A and B. Note that components with smaller $k$-radius cannot be arranged to be hexagonal (not even approximately), so they will have a low gridness score. In contrast, the next components with higher $k$-radius (e.g., group C) can form an approximately hexagonal shape together, and would appear as an additional grid cell 'module'. The grid spacing of this new module will decrease by $\sqrt{2}$, since the new $k$-radius is about $\sqrt{2}$ times larger than the $k$-radius of groups A and B.

## Acknowledgements

We would like to thank Alexander Mathis and Omri Barak for comments on the manuscript, and Gilad Tocker and Matan Sela for helpful discussions and advice. The research was supported by the Israel Science Foundation grants 955/13 and 1882/13, by a Rappaport Institute grant, and by the Allen and Jewel Prince Center for Neurodegenerative Disorders of the Brain. The work of RM and Y D was partially supported by the Ollendorff Center of the Department of Electrical Engineering, Technion. DD is a David and Inez Myers Career Advancement Chair in Life Sciences fellow. The work of DS was partially supported by the Gruss Lipper Charitable Foundation, and by the Intelligence

Advanced Research Projects Activity (IARPA) via Department of Interior/Interior Business Center (DoI/IBC) contract number D16PC00003. The U.S. Government is authorized to reproduce and distribute reprints for Governmental purposes notwithstanding any copyright annotation thereon. Disclaimer: The views and conclusions contained herein are those of the authors and should not be interpreted as necessarily representing the official policies or endorsements, either expressed or implied, of IARPA, DoI/IBC, or the U.S. Government.

## Additional information

### Funding

| Funder | Grant reference number | Author |
|---|---|---|
| Ollendroff center of the Department of Electrical Engineering, Technion | Research fund | Yedidyah Dordek Ron Meir |
| Gruss Lipper Charitable Foundation | | Daniel Soudry |
| Intelligence Advanced Research Projects Activity | | Daniel Soudry |
| Israel Science Foundation | Personal Research Grant, 955/13 | Dori Derdikman |
| Israel Science Foundation | New Faculty Equipment Grant, 1882/13 | Dori Derdikman |
| Rappaport Institute | Personal Research Grant | Dori Derdikman |
| Allen and Jewel Prince Center for Neurodegenrative Disorders | Research Grant | Dori Derdikman |

The funders had no role in study design, data collection and interpretation, or the decision to submit the work for publication.

### Author contributions

YD, RM, DD, Conception and design, Analysis and interpretation of data, Drafting or revising the article; DS, Conception and design, Analysis and interpretation of data, Drafting or revising the article, Analytical derivations in Materials and methods and in Appendix

### Author ORCIDs

Dori Derdikman, http://orcid.org/0000-0003-3677-6321

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

## Appendix

### Movement schema of agent and environment data

The relevant simulation data used:

| Size of arena 10X10 | Place cells field width: 0.75 | Place cells distribution: uniform |
|---|---|---|
| Velocity: 0.25 (linear), 0.1-6.3 (angular) | # Place cells: 625 | Learning rate: 1/(t+1e5) |

The agent was moved around the virtual environment according to:

$$D^{t+1} = \mathrm{modulo}(D^t + \omega \cdot Z, 2\pi)$$
$$x^{t+1} = x^t + \nu \cdot \cos(D^{t+1})$$
$$y^{t+1} = y^t + \nu \cdot \sin(D^{t+1})$$

where $D^t$ is the current direction angle, $\omega$ is the angular velocity, $Z - N(0,1)$ where $N$ is the standard normal Gaussian distribution, $\nu$ is the linear velocity, and $(x^t, y^t)$ is the current position of the agent. Edges were treated as periodic – when agent arrived to one side of box it was teleported to the other side.

### Positive-negative disks

Positive-negative disks are used with the following activity rules:

$$\left( r_j^t(x^t, y^t) \right) = \begin{cases} \frac{1}{\rho_1^2}, & \sqrt{(x^t - c_{x,j})^2 + (y^t - c_{y,j})^2} < \rho_1 \\ \frac{1}{\rho_2^2 - \rho_1^2}, & \rho_1 < \sqrt{(x^t - c_{x,j})^2 + (y^t - c_{y,j})^2} < \rho_2 \\ 0, & \rho_2 < \sqrt{(x^t - c_{x,j})^2 + (y^t - c_{y,j})^2} \end{cases}$$

Where $c_x$, $c_y$ are the centers of the disks, and $\rho_1$, $\rho_2$ are the radii of the inner circle and the outer ring, respectively. The constant value in the negative ring was chosen to yield zero integral over the disk.

### Fourier transform of the difference of Gaussians function

Here we prove that if we are given a difference of Gaussians function,

$$r(\mathbf{x}) = \sum_{i=1}^{2} (-1)^{i+1} c_i \exp\left( -\frac{\mathbf{x} \cdot \mathbf{x}}{2\sigma_i^2} \right),$$

in which the appropriate normalization constants for a bounded box are

$$c_i = \left[ \int_{-L}^{L} \exp\left( -\frac{z^2}{2\sigma_i^2} \right) dz \right]^{-1} = \left[ \sqrt{2\pi\sigma_i^2} [2F(L/\sigma_i) - 1] \right]^{-D},$$

with $F(x) = \frac{1}{\sqrt{2\pi\sigma_i^2}} \int_{-\infty}^{x} \exp\left( -\frac{z^2}{2\sigma_i^2} \right) dz$ being the cumulative normal distribution, then

$\forall \mathbf{k} \in \hat{S} = \left\{ \left( \frac{m_1 2\pi}{L}, \ldots, \frac{m_D 2\pi}{L} \right) \right\}_{(m_1, \ldots, m_D) \in \mathbb{Z}^D}$, we have

$$\hat{r}(\mathbf{k}) \overset{\Delta}{=} \frac{1}{|S|} \int_S r(\mathbf{x}) \, e^{i\mathbf{k}\cdot\mathbf{x}} \, d\mathbf{x} = \frac{1}{|S|} \sum_{i=1}^{2} (-1)^{i+1} \exp\left(-\frac{1}{2}\sigma_i^2 \, \mathbf{k}\cdot\mathbf{k}\right).$$

Note that the normalization constants $c_i$ vanish after the Fourier transform, and that in the limit $L \gg \sigma_1, \sigma_2$ we obtain the standard result for the Fourier transform of an unbounded Gaussian distribution.

Proof: For simplicity of notation, we assume $D = 2$. However, the calculation is identical for any $D$.

$$
\begin{aligned}
\hat{r}(\mathbf{k}) &= \frac{1}{|S|} \int_S r(\mathbf{x}) e^{i\mathbf{k}\cdot\mathbf{x}} d\mathbf{x} \\[6pt]
&= \frac{1}{|S|} \sum_{i=1}^{2} \int_{-L}^{L} dx \int_{-L}^{L} dy \left[ (-1)^{i+1} c_i \exp\left(-\frac{x^2+y^2}{2\sigma_i^2}\right) \right] \exp\left(i(k_x x + k_y y)\right) \\[6pt]
&= \frac{1}{|S|} \sum_{i=1}^{2} \left[ \int_{-L}^{L} dx \left[ (-1)^{i+1} c_i \exp\left(-\frac{x^2}{2\sigma_i^2}\right) \right] \exp(ik_x x) \right] \\[6pt]
&\qquad\cdot \left[ \int_{-L}^{L} dy \left[ (-1)^{i+1} \exp\left(-\frac{y^2}{2\sigma_i^2}\right) \right] \exp(ik_y y) \right] \\[6pt]
&= \frac{2^D}{|S|} \sum_{i=1}^{2} (-1)^{i+1} c_i \prod_{z\in\{x,y\}} \int_0^L dz \exp\left(-\frac{z^2}{2\sigma_i^2}\right) \left[\exp(ik_z z) + \exp(-ik_z z)\right] \\[6pt]
&= \frac{2^D}{|S|} \sum_{i=1}^{2} (-1)^{i+1} c_i \prod_{z\in\{x,y\}} \int_0^L dz \exp\left(-\frac{z^2}{2\sigma_i^2}\right) \cos(k_z z)
\end{aligned}
$$

To solve this integral, we define

$$I_i(a) = \int_0^L dz \exp\left(-\frac{z^2}{2\sigma_i^2}\right) \cos(az).$$

Its derivative is

$$
\begin{aligned}
I'_i(a) &= -\int_0^L dz\, z \exp\left(-\frac{z^2}{2\sigma_i^2}\right) \sin(az) \\[6pt]
&= \sigma_i^2 \exp\left(-\frac{z^2}{2\sigma_i^2}\right) \sin(az)\Big|_{z=0}^{L} - a\sigma_i^2 \int_0^L dz \exp\left(-\frac{z^2}{2\sigma_i^2}\right) \cos(az) \\[6pt]
&= -a\sigma_i^2 I_i(a)
\end{aligned}
$$

where in the last equality we used the fact that $aL = 2\pi n$ (where $n$ is an integer) if $a \in \hat{S}$. Solving this differential equation, we obtain

$$I_i(a) = I_i(0) \exp\left(-\frac{1}{2}\sigma_i^2 a^2\right)$$

where

$$I_i(0) = \int_0^L dz \exp\left(-\frac{z^2}{2\sigma_i^2}\right) = \sqrt{2\pi\sigma_i^2}\left[F\left(\frac{L}{\sigma_i}\right) - F(0)\right] = \sqrt{2\pi\sigma_i^2}\left[F\left(\frac{L}{\sigma_i}\right) - \frac{1}{2}\right].$$

substituting this into our last expression for $r(\mathbf{k})$ we obtain

$$r(\mathbf{k}) = \frac{1}{|S|} \sum_{i=1}^{2} (-1)^{i+1} \exp\left(-\frac{1}{2}\sigma_i^2 \sum_{z \in \{x,y\}} k_z^2\right),$$

which is what we wanted to prove.

## Why the 'unconstrained' $k_\dagger$ is a lower bound on 'constrained' $k_*$

In section 'The PCA solution with a non-negative constraint – 1D Solutions', we mention that the main difference between the Equations (**Equations 33 and 34**) is the base frequency, $k_*$, which is slightly different from $k_\dagger$.

Here we explain how the relation between $k_*$ and $k_\dagger$ depends on the $k$-lattice discretization, as well as the properties of $\hat{r}(k)$. The discretization effect is similar to the unconstrained case, and can cause a difference of at most $\pi/L$ between $k_*$ to $k_\dagger$. However, even if $L \longrightarrow \infty$, we can expect $k_*$ to be slightly smaller than $k_\dagger$. To see that, suppose we have a solution as described above, with $k_* = k_\dagger + \delta k$ and $\delta k$ is a small perturbation (which does not affect the non-negativity constraint). We write the perturbed $k$-space objective of this solution

$$\sum_{m=-\infty}^{\infty} |\hat{J}(mk_\dagger)|^2 |\hat{r}\left(m(k_\dagger + \delta k)\right)|^2$$

Since $k_\dagger$ is the peak of $|\hat{r}(k)|^2$, if $|\hat{r}(k)|^2$ is monotonically decreasing for $k > k_\dagger$ (as is the case for the difference of Gaussians function), then any positive perturbation $\delta k > 0$ would decrease the objective.

However, a sufficiently small negative perturbation $\delta k < 0$ would improve the objective. We can see this from the Taylor expansion of the objective,

$$\sum_{m=-\infty}^{\infty} |\hat{J}(mk_\dagger)|^2 \left(|\hat{r}(mk_\dagger)|^2 + \frac{d|\hat{r}(k)|^2}{dk}\Big|_{k=mk_\dagger} m\delta k + O(\delta k)^2\right)$$

In which the derivative $\frac{d|\hat{r}(k)|^2}{dk}\Big|_{k=mk_\dagger}$ is zero for $m = 1$ (since $k_\dagger$ is the peak of $|\hat{r}(k)|^2$) and negative for $m > 1$, if $|\hat{r}(k)|^2$ is monotonically decreasing for $k > k_\dagger$. If we gradually increase the magnitude of this negative perturbation $\delta k < 0$, at some point the objective will stop increasing and start decreasing, because, for the difference of Gaussians function, $|\hat{r}(k)|^2$ increases more sharply for $k < k_\dagger$ than it decreases for $k > k_\dagger$. We can thus treat the 'unconstrained' $k_\dagger$ as an upper bound for the 'constrained' $k_*$, if we ignore discretization effects (*i.e.*, the limit $L \longrightarrow \infty$).

## Upper bound on the constrained objective – 2D square lattice

In the Methods section 'The PCA solution with a non-negative constraint – 2D Solutions' we examine different 2D lattice-based solutions, including a square lattice, which is a solution of the following constrained minimization problem

$$\max_{\{J_m\}_{m=0}^{\infty}} \hat{J}_{1,0}^2 + \hat{J}_{0,1}^2$$

$$\text{s.t.:} \quad (1) \sum_{m_x=-\infty}^{\infty} \sum_{m_y=-\infty}^{\infty} \hat{J}_{m_x,m_y}^2 = 1$$

$$(2) \sum_{m_x=-\infty}^{\infty} \sum_{m_y=-\infty}^{\infty} \hat{J}_{m_x,m_y} \cos\left( k_\dagger (m_x x + m_y y) + \phi_{m_x,m_y} \right) \geq 0$$

Here we prove that the objective is bounded above by $\frac{1}{4}$, i.e., $\hat{J}_{1,0}^2 + \hat{J}_{0,1}^2 \leq \frac{1}{4}$.

Without loss of generality we assume $\phi_{1,0} = \phi_{0,1} = 0, k_\dagger = 1$ (we can always shift and scale $x, y$). We denote

$$P(x,y) = 2\hat{J}_{0,1}\cos(x) + 2\hat{J}_{1,0}\cos(y)$$

and

$$A(x,y) = \sum_{m_x=-\infty}^{\infty} \sum_{m_y=-\infty}^{\infty} \hat{J}_{m_x,m_y} \cos\left( (m_x x + m_y y) + \phi_{m_x,m_y} \right) - P(x,y)$$

We examine the domain $[0, 2\pi]^2$. We denote by $C^\pm$ the regions in which $P(x,y)$ is positive or negative, respectively. Note that

$$P\left( (x+\pi)\bmod 2\pi, (y+\pi)\bmod 2\pi \right) = -P(x,y)$$

so both regions have the same area, and

$$\int_{C^-} P^2(x,y)dxdy = \int_{C^+} P^2(x,y)dxdy. \tag{A1}$$

From the non-negativity constraint (2), we must have

$$\forall (x,y) \in C^- : A(x,y) \geq -P(x,y). \tag{A2}$$

Also, note that $P(x,y)$ and $A(x,y)$ do not share any common Fourier (cosine) components. Therefore, they are orthogonal, with the inner product being the integral of their product on the region $[0, 2\pi]^2$

$$\begin{aligned}
0 &= \int_{[0,2\pi]^2} P(x,y)A(x,y)dxdy \\
&= \int_{C^+} P(x,y)A(x,y)dxdy + \int_{C^-} P(x,y)A(x,y)dxdy, \\
&\leq \int_{C^+} P(x,y)A(x,y)dxdy - \int_{C^+} P^2(x,y)dxdy
\end{aligned}$$

where in the last line we used the bound from **Equation (A2)**, and then **Equation (A1)**. Using this result, together with Cauchy-Schwartz inequality, we have

$$\int_{C^+} P^2(x,y)dxdy \leq \int_{C^+} P(x,y)A(x,y)dxdy \leq \sqrt{\int_{C^+} P^2(x,y)dxdy \int_{C^+} A^2(x,y)dxdy},$$

So,

$$\int_{C^+} P^2(x,y)\,dxdy \le \int_{C^+} A^2(x,y)\,dxdy,$$

Summing this equation with **Equation (A2)**, squared and integrated over the region $C^-$, and dividing by $(2\pi)^2$, we obtain

$$\frac{1}{(2\pi)^2}\int_{[0,2\pi]^2} P^2(x,y)\,dxdy \le \frac{1}{(2\pi)^2}\int_{[0,2\pi]^2} A^2(x,y)\,dxdy,$$

using the orthogonality of the Fourier components (*i.e.*, Parseval's theorem) to perform the integrals over $P^2(x,y)$ and $A^2(x,y)$, we get

$$2\hat{J}_{1,0}^2 + 2\hat{J}_{0,1}^2 \le \sum_{m_x=-\infty}^{\infty}\sum_{m_y=-\infty}^{\infty}\hat{J}_{m_x,m_y}^2 - 2\hat{J}_{1,0}^2 - 2\hat{J}_{0,1}^2.$$

Lastly, plugging in the normalization constraint $\sum_{m_x=-\infty}^{\infty}\sum_{m_y=-\infty}^{\infty}\hat{J}_{m_x,m_y}^2 = 1$, we find

$$\frac{1}{4} \ge \hat{J}_{1,0}^2 + \hat{J}_{0,1}^2$$

as required.

