## [Decision Letter]

Thank you for submitting your work entitled "Extracting grid characteristics from spatially distributed place cell inputs using non-negative PCA" for peer review at *eLife*. Your submission has been favorably evaluated by Timothy Behrens (Senior editor), Reviewing editor Michael Frank, and two reviewers.

The reviewers have discussed the reviews with one another and the Reviewing editor has drafted this decision to help you prepare a revised submission.

Summary:

The authors present analytical and neural network simulations which demonstrate that, under certain constraints, the first principal component of place cell firing patterns in a 2D arena resemble hexagonal grid cell firing patterns. Specifically, these constraints require that the principal components be constrained to non-negativity, and that the place cell inputs have a zero mean in time or space. In addition, they suggest that interesting features of grid cell firing, namely the tendency for grid scale to be discretely clustered, and for the grid to align at approximately 7 degrees to the walls of a square arena. This is an important result which adds to a growing body of recent studies that have emphasised the importance of the projection from place cells to grid cells, as opposed to the grid cell to place cell projection that has been the subject of several previous theoretical studies.

As you will see however, the Reviewers felt that it was important to provide some intuition and cogent explanation about how the results were obtained, and several questions arose along these lines. It is not clear how the dual constraints of (eigenvector) non-negativity and orthogonality are actually satisfied, nor is it clear why they get 7degree alignment or a ratio of 1.4 for the jump in grid scale.

In addition to the revisions articulated below, the following suggestion arose from discussion amongst the Reviewers/editors.

Figure 13 of the manuscript tracks the evolution of the network from different initial conditions.

One might go some way in clarifying the questions of how and why the observed phenomena occur by tracking individual objective measures, in a manner similar to Figure 13.

To wit, one of the algorithms you use trades off two objectives:

(i) maximize the variance of the projection of the data onto the [non-negative] principal component

(ii) enforce approximate orthogonormality by minimizing || I – e^T^ e||, where I is the identity matrix, and e is the normalized "eigenvector," subject to the non-negativity constraint.

In addition, one ought to quantify the degree of non-negativity. There are several ways one might go about this, but one approach would be to normalize the eigenvector to unit length, then take the norm of the vector composed of only the positive entries and subtract the norm of the vector of negative entries.

Essential revisions:

1) This manuscript is of sufficient general interest for publication in *eLife*, although there is a general lack of detail when describing the methodology. There is also significant overlap with a recent paper at NIPS 2014 (Stachenfeld et al., 2014). I find both papers (Dordek et al., submitted, and Stachenfeld et al.) extremely interesting, and they both have different focusses. But I do think that there needs to be some link to/discussion of Stachenfeld et al.

2) Perhaps the most important issue is that they provide little intuition into the interesting subsidiary results. There is a section on "Why Hexagons?" but none on why other aspects of the results occur. This becomes more important given the fact that Stachenfeld also see the basic result. Why do grid scales have a ratio of 1.4 (indeed is this figure reliable)? Why do they see a non-zero alignment of grids to borders (and is this figure reliable)? Alignment angle appears to depend on place field size, but how does overall place cell firing coverage (and covariance) vary with place field size? How does the alignment angle covary with gridness (the very nice grid-like patterns they show tend to look aligned)?

3) "Due to the isotropic nature of the data (generated by the agent's motion), there was a 2D redundancy in the X-Y plane in conjunction with a dual phase redundancy in 1D, resulting in a fourfold total redundancy of every solution" – this statement requires unpacking. Presumably the dual phase redundancy is caused by the periodic boundary conditions? What are the implications for non-periodic boundary conditions? This sentence needs to be explained to the non-expert reader.

4) Similarly, Figure 4 is cryptic – how does the figure show fourfold redundancy? The authors also use a reduced number of place cells to obtain this result (225 vs. 625 in other simulations) – what are the reasons for this change? This should be explained and justified in the text.

5) More generally, when changing place field size, what happens to the number of place cells – presumably the density of firing overlap (covariance) needs to be maintained as this will strongly affect the results?

6) It is not clear how the non-negative weights are implemented alongside zero mean inputs, and we could not gain any additional insight from the Methods section. This issue needs to be elaborated upon in the text, otherwise it seems implicitly contradictory. Does it imply that the output GC can have negative firing rates? i.e. are there positive and negative place cell firing rates (with a mean of zero), coupled with positive place cell to grid cell synaptic weights, to give both negative and positive inputs to the output grid cell? Presumably the time step to time step changes in grid cell firing rate between negative and positive values will have a large impact on the direction of synaptic weight change – is this the case?

7) Figure 7; also 14B – the analytical PCA results for constrained weights appear to exhibit a bimodal distribution of 90 degree gridness scores – what in the data accounts for this? Can the authors provide any comment or insight?

"The dependency was almost linear" – presumably there is a mathematical reason for this, based on the relationship between the covariance matrix and eigenvectors? Can the authors provide any comment or insight into this relationship?

8) The methods are generally lacking in detail that would allow these simulations to be replicated, based on details provided in the manuscript (see specific details below). The authors should ensure that sufficient detail to allow replication is incorporated. They may also like to consider uploading their code to ModelDB, or a similar online repository, for the sake of transparency and to allow independent replication.

9) The representation of space lies on a torus in this model, which means that the eigenvectors should be periodic and the spatial frequencies quantized. We fail to grasp why, in Figure 3, the first spatial frequency should be 2 and not 1 (i.e., we have two peaks in each direction), and why the eigenvectors 3B is periodic, but in 3A they are not.

10) The eigenvectors of a correlation matrix are orthogonal. If we normalize these eigenvectors and furthermore enforce an additional constraint that the entries of each eigenvector must be positive semidefinite, then the "synaptic weights" of different eigenvectors should not overlap. In other words, if an entry in the first eigenvector is non-zero, it ought to be zero in all other eigenvectors.

11) While the algorithms the authors use relax this last constraint, it is not clear how skewing or shearing a square pattern that results from PCA helps satisfy the dual constraints of non-negativity and orthogonality.

This is a key point in the paper, and begs for an intuitive explanation, or at least more analysis or quantification of the degree to which the constraints are satisfied.

12) The finding that the spacings corresponding to successive eigenvectors obey ~ 2 also calls for an explanation. Why should this ratio of spacings best satisfy non-negativity, orthogonality, and periodicity?

[Editors' note: further revisions were requested prior to acceptance, as described below.]

Thank you for resubmitting your work entitled "Extracting grid characteristics from spatially distributed place cell inputs using non-negative PCA" for further consideration at *eLife*. Your revised article has been favorably evaluated by Timothy Behrens (Senior editor), Reviewing editor Michael Frank, and two reviewers, one of whom is Neil Burgess. The manuscript has been improved but both reviewers have identified some remaining issues that need to be addressed before acceptance, as articulated in the comments below:

*Reviewer #1:*

The authors have worked hard to provide further intuition and interpretation to their results, with much success, although there are still some points that should be clarified, see comments below.

There is also a plain error that should be removed. The authors state "the distribution of grid orientations becomes uniform in the range of 0-15 degrees, with a mean (and median) at 7.5 degrees, similar to experimental data [Stensola et al., 2015]". On the contrary the experimental data shows a very non-uniform distribution clearly peaked at about 7.5 degrees, which is quite different. Also, as a circular variable limited to [0,15) – albeit one for which 16 degrees corresponds to 14 degrees rather than 1 degree – the 'mean' and 'median' they refer to are undefined if the distribution is uniform. Thus the authors should acknowledge that they do not provide an explanation for this aspect of the experimental data, and some of the text/ results on this point could be removed, most importantly in the Abstract.

At the start of describing the main result (Results, first paragraph) is not clear that the "1d mapping of the 2d activity" is caused by the trajectory of the rat moving through the place fields.

The next part of the description of their PCA analysis as Eigenvectors and their spatial interpretation could also be improved in terms of intuition. It would be worth mentioning that the PCA, in addition to being calculated as the Eigenvectors of the matrix of temporal covariance between inputs, can also be thought of in terms of singular value decomposition of the (appropriately normalised) input activity. I.e. the input to the network (temporal sequences of place cell activity) can be decomposed into an outer product of (spatial) weights over place cell inputs (their Eigenvectors) and temporally varying weights over these Eigenvectors. The network learns the spatial weights over place cells (the Eigenvectors) as the connections weights from the place cells, and "projection onto place cell space" is simply the firing rates of the output neuron plotted against the location of the rat.

In the second paragraph of the subsection “Adding a non-Negativity constraint to the PCA “it is important to remind the reader that the "raw place cells activity" is not the final input to the PCA numerical methods – spatial or temporal mean normalization has to happen first.

It would be useful to have a surface plot showing Gridness vs. PC Density vs. PC Width. I have found it hard to replicate the Gridness distribution and wonder how general it is to the precise place cell inputs used? The reference to Figure 10 in the third paragraph of the subsection “Effect of place cell parameters on grid structure “is incorrect (I think) – not showing the dependence on the width of the place fields.

Regarding the presence of "modules" of grid cells (subsection “Modules of grid cells”), it is not clear if the interpretation is that different modules correspond to different grid scales, or different Eigenvectors (even if having the same scale). Assuming the latter, it seems that the analysis predicts that there should only be two modules, with the scale of the first module approximately 7.5x the largest place field inputs, and the second module (~2 smaller) existing only to enable non-negative PCA solutions. This deserves discussion, given that the data show at least 5 modules, with a range of scales, the smallest and most numerous having approximately the scale of the smaller place fields found in the dorsal hippocampus (25-30 cm).

*Reviewer #2:*

The new material, starting with the subsection “Steady state analysis “is great, and just the thing I was asking for, namely an understanding as to *why* hexagonal patterns result under a non-negative PCA learning rule.

There are a few things I'd like to understand better, though:

The last equation in the section “2D Solutions”: should read *ϕ_1_ = ϕ_2_ = ϕ_3_*n 2 π/3, n ϵ Z. e.g. *ϕ_1_ = ϕ_2_ = ϕ_3_*= 0 would be a solution.

As written, *ϕ_1_ = ϕ_2_ = ϕ_3_*, is not the maximum. Besides, this more general case would allow solutions that are unlike the ones observed in experiments.

This brings me to

In the subsection “2D solutions”:

"To do this [maximize the Fourier components inside the basis set,] we must maximize the minimal value of the cosine components for the basis set"

I am afraid this argument escapes me, even though its role is crucial.

While I think I understand the Fourier domain argument up to this point in the text, how exactly does maximizing the minimum value in real space of the function guarantee that the Fourier components in Eq. 24 are maximized?

Eq. 34: why is the upper limit here ∞, when it was *M* in Eq. 33?

I'd like some clarification from the authors before I can make a judgment call as to whether the argument is likely to be correct.

---

## [Author Response]

*As you will see however, the Reviewers felt that it was important to provide some intuition and cogent explanation about how the results were obtained, and several questions arose along these lines. It is not clear how the dual constraints of (eigenvector) non-negativity and orthogonality are actually satisfied, nor is it clear why they get 7degree alignment or a ratio of 1.4 for the jump in grid scale.* Re-analysis demonstrates that we tend to get values which are on average a little lower than 7 degrees (Author response Figure 3 and Figure 4). However, the range of possible values is from 0 to 15 degrees, and we demonstrate analytically (in the new extended Methods section) that in the case of a very large box, the distribution of orientation angles becomes uniform in the interval 0-15, with a mean at 7.5 degrees, consistent with experimental data.

The jump of 1.4 results from the change in the circular components of the solution in Fourier space (See especially new Figure 17 in Methods section of paper). From running more simulations and also from analytical considerations, we believe that within our theoretical framework there is only one jump, and thus this model can really only explain 2 modules and not more (see analytical derivation for more intuition on this point).

In addition to the new extended Methods section, we have updated the Abstract and Results (throughout) to account for these changes.

In addition to the revisions articulated below, the following suggestion arose from discussion amongst the Reviewers/editors. […]

*In addition, one ought to quantify the degree of non-negativity. There are several ways one might go about this, but one approach would be to normalize the eigenvector to unit length, then take the norm of the vector composed of only the positive entries and subtract the norm of the vector of negative entries.*

We thank the editors and reviewers for raising this essential issue. In our understanding, the main issue is to comprehend to what extent we can combine maximal -variance, orthogonality, and positivity together, when looking at multiple outputs (such as in old Figure 11 and Figure 12, or new Figure 13 and Figure 14). As we demonstrate now, the outputs from the method are not orthogonal (Figure 18), and this we believe solves the conundrum. We now added a short comment about this into the Results section: “It is important to note though that, due to the non-negativity constraint, the vectors achieved in this way were not orthogonal, and thus it cannot be considered a real orthogonalization process, although as explained in the Methods section, the process does aim for maximum difference between the vectors”

Author response image 1.Demonstrates that output vectors are not orthogonal.Matrix of cosine values between every two vectors.**DOI:**
http://dx.doi.org/10.7554/eLife.10094.023

*Essential revisions: 1) This manuscript is of sufficient general interest for publication in eLife, although there is a general lack of detail when describing the methodology.*

Additional detail has been added to the Methods. Furthermore, extensive sections have been added to the Methods in order to provide an analytical motivation to the whole paper (from the subsection “Steady state analysis”). See detailed points below in the section related to Methods. Furthermore, we have uploaded our MATLAB code to a public server, as described below.

*There is also significant overlap with a recent paper at NIPS 2014 (Stachenfeld et al., 2014). I find both papers (Dordek et al., submitted, and Stachenfeld et al) extremely interesting, and they both have different focusses. But I do think that there needs to be some link to/discussion of Stachenfeld et al.*

Stachnfeld et al. arrive to similar ideas in a very nice paper based on a reinforcement learning model of place cells. Unlike our paper, their paper does not contain the positivity constraint, and thus the “grid cells” that results from the PCA procedure they use are square rather than hexagonal. Furthermore, they did not provide a theoretical analysis of the nature provided in the revision. We now discuss their paper in the Discussion, as follows: “We note that similar work has noticed the relation between place-cell-to-grid-cell transformation and PCA.[…] However, due to the unconstrained nature of their transformation, the outputted grid cells were square-like.”

*2) Perhaps the most important issue is that they provide little intuition into the interesting subsidiary results. There is a section on "Why Hexagons?" but none on why other aspects of the results occur. This becomes more important given the fact that Stachenfeld also see the basic result. Why do grid scales have a ratio of 1.4 (indeed is this figure reliable)?*

Following the new analytical derivation in the Methods section (from the subsection “Steady state analysis”), we can now explain the jump of scale as a hop on the Fourier-space lattice (new Figure 17 in paper).

We have now re-run the multiple solutions in order to demonstrate a jump between two consecutive scales. However, we find that it is harder to see the jump to the next smaller third scale, and thus we conclude that the method only partially accounts for the effect of modules. This is now noted in the Results as follows: “[…]we found that the ratio between the distances of the modules was ~1.4, close to the value of 1.42 found by Stensola et al. [Stensola et al., 2012]. Although we searched for additional such jumps, we could only identify this single jump, suggesting that our model can yield up to two “modules” and not more.”

*Why do they see a non-zero alignment of grids to borders (and is this figure reliable)?*

As we now show in our analytical derivation, the non-zero alignment can arise from the relation between the lattice points in the Fourier domain and the optimal-solution circle. This is now explainedthoroughly in the new sections added to the Methods from the subsection “Steady state analysis” and in Figure 15–Figure 16 in paper.

*Alignment angle appears to depend on place field size, but how does overall place cell firing coverage (and covariance) vary with place field size?*

In our previous submission we manipulated the place field size while keeping the size of the arena and number of place cells constant. Thus the coverage has increased as a function of the increase in size. Here (Figure 19) we show an example of increasing both the place field size and the size of the arena, while keeping the place-cell-size/arena size constant, such that the total coverage increases (a place cell on every pixel).

*How does the alignment angle covary with gridness (the very nice grid-like patterns they show tend to look aligned)?*

In Figure 19 we plot grid orientation as a function of gridness. As can be seen the two quantities are not strongly related:

Author response image 2.Example of changing width of place cells (σ) and correspondingly changing arena size, such that there is a fixed ratio: arena_size/σ = 40.(FISTA algorithm was used for positive PCA, with 0 boundary conditions.) (**a**) Grid scale as a function of σ (place cell width); (**b**) Grid orientation as a function of σ. (**c**) Histogram of grid orientations (**d**) Mean gridness as a function of σ. (**e**) Histogram of mean gridness.**DOI:**
http://dx.doi.org/10.7554/eLife.10094.024

Author response image 3.Effect of changing the place-field size in a constant boundary (FISTA algorithm used; zero boundary conditions and Arena size 500);(**A**) Grid scale as a function of place field size (σ); Linear fit is: Scale = 7.4. Σ+0.62; the lower bound is equal to , where k† is defined in Eq. 32 in the Methods section; (**B**) Grid orientation as a function of gridness (green line is the mean); (**C**) Grid orientation as a function of σ – scatter plot (blue stars) and mean (green line); (**D**) Histogram of grid orientations; (**E**) Mean gridness as a function of σ; and (**F**) Histogram of mean gridness.**DOI:**
http://dx.doi.org/10.7554/eLife.10094.025

*3) "Due to the isotropic nature of the data (generated by the agent's motion), there was a 2D redundancy in the X-Y plane in conjunction with a dual phase redundancy in 1D, resulting in a fourfold total redundancy of every solution" – this statement requires unpacking. Presumably the dual phase redundancy is caused by the periodic boundary conditions?*

We have now devoted a large section in the Methods to analytical considerations, which can explain the fourfold redundancy. Thus we changed the wording of the above paragraph to the following: “The solution demonstrated a fourfold redundancy (Figure 4). This was apparent in the plotted eigenvalues (from largest to the smallest eigenvalue, Figure 4), which demonstrated a fourfold grouping-pattern. The fourfold redundancy can be explained analytically by the symmetries of the system – see PCA analysis in Methods section.”

*What are the implications for non-periodic boundary conditions? This sentence needs to be explained to the non-expert reader.*

In the case of very large arenas, it does not matter what the boundary conditions are. In smaller arenas, the nature of the boundary conditions (i.e. how the place cells behave at the boundary) has an effect. It seems from our simulations that the boundary conditions matter less for the scale of the grid and more for the grid orientation (Compare Author response Figure 3 and Figure 4). We now note in the Results section: “[…]for very large environments the effects of boundary conditions diminish.”.

Author response image 4.Effect of changing the place-field size in a constant boundary (FISTA algorithm used; periodic boundary conditions and Arena size 500);(**A**) Grid scale as a function of place field size (σ); Linear fit is: Scale = 7.4 Σ+0.62; the lower bound, equal to 2π/k†, were k† is defined in Eq. 32 in the Methods section; (**B**) Grid orientation as a function of gridness; (**C**) Grid orientation as a function of σ – scatter plot (blue stars) and mean (green line); (**D**) Histogram of grid orientations; (**E**) Mean gridness as a function of σ; and (**F**) Histogram of mean gridness.**DOI:**
http://dx.doi.org/10.7554/eLife.10094.026

</Figure 21 title/legend>

*4) Similarly, Figure 4 is cryptic – how does the figure show fourfold redundancy? The authors also use a reduced number of place cells to obtain this result (225 vs. 625 in other simulations) – what are the reasons for this change? This should be explained and justified in the text.* We have improved the figure by increasing the resolution and using the same number of points for both sub-figures. Now the fourfold redundancy is quite clear (Figure 4). We updated Figure 4 to contain all 625 values, and have also added an additional panel to emphasize the fourfold redundancy.

*5) More generally, when changing place field size, what happens to the number of place cells – presumably the density of firing overlap (covariance) needs to be maintained as this will strongly affect the results?*

Figure 19 above demonstrates that results are not strongly dependent on the place cell overlap.

*6) It is not clear how the non-negative weights are implemented alongside zero mean inputs, and we could not gain any additional insight from the Methods section. This issue needs to be elaborated upon in the text, otherwise it seems implicitly contradictory. Does it imply that the output GC can have negative firing rates? i.e. are there positive and negative place cell firing rates (with a mean of zero), coupled with positive place cell to grid cell synaptic weights, to give both negative and positive inputs to the output grid cell?*

The grid cells were not clipped to zero so they did obtain negative values. In order to realize this in real neurons we will need to assume a general baseline firing which will account for the fact that real neurons do not have negative firing rates.

*Presumably the time step to time step changes in grid cell firing rate between negative and positive values will have a large impact on the direction of synaptic weight change – is this the case?*

Yes, this is indeed the case, as far as we understand the comment.

*7) Figure 7; also 14B – the analytical PCA results for constrained weights appear to exhibit a bimodal distribution of 90 degree gridness scores – what in the data accounts for this? Can the authors provide any comment or insight?*

The bi-modal distribution was a result of the formation of two populations, one with small orientations to one of the walls and another with small orientation to the other wall. We believe this is not an essential result, but cannot provide a full explanation for this phenomenon.

*"The dependency was almost linear" – presumably there is a mathematical reason for this, based on the relationship between the covariance matrix and eigenvectors? Can the authors provide any comment or insight into this relationship?*

This relation is now explained from the added analytical model. The predicted value provides a tight lower bound on the actual relation, as can be seen from the results of simulations in Figure 20 and Figure 21 above, where the actual values were that the grid scale is about 7.5 times the width of the place field.

Intuitively, the linear dependency between the place cell width and the grid cell spacing must be true in the limit of infinite box size, from dimensional analysis: the length units of the grid cell spacing must be the same length units of the place cell width, since these are the only length units in the model. This is explained mathematically in the unconstrained case by Equation 32, as we detail below that equation:

"Notice that if we multiply the place cell width by some positive constant c, then the solution 2pi/k† will be divided by c. […] When the box has finite size, k-lattice discretization also has a (usually small) effect on the grid spacing."

A similar reasoning can be applied to the constrained case.

*8) The methods are generally lacking in detail that would allow these simulations to be replicated, based on details provided in the manuscript (see specific details below). The authors should ensure that sufficient detail to allow replication is incorporated. They may also like to consider uploading their code to ModelDB, or a similar online repository, for the sake of transparency and to allow independent replication.*

We have followed the advice given and have now uploaded our code to t

This is also noted at the beginning of the Methods section.

*9) The representation of space lies on a torus in this model, which means that the eigenvectors should be periodic and the spatial frequencies quantized. We fail to grasp why, in Figure 3, the first spatial frequency should be 2 and not 1 (i.e., we have two peaks in each direction),*

Indeed the spatial frequencies are quantized by the box dimensions. However, the eigenvectors periodicity is not determined by the box, as we now explain mathematically in extended Methods section. Even in the case that the box size is infinite, we get a finite periodic solution, with spatial frequency equal to the peak of the Fourier transform of the place cell tuning curve – given in Equation 32. When the box size is finite this spatial frequency is also affected by the quantization due to the box dimensions (Figure 15).

*and why the eigenvectors 3B is periodic, but in 3A they are not.*

In Figure 3 we show multiple examples of the 1^st^ principal component as output from the network, while in Figure 3 we show the first 16 components of the PCA algorithm. Thus both figures do not show the same thing. Generally, the PCA components need not appear periodic (i.e., repeat more than once in some directions), though they do obey periodic boundary conditions. For example, PCA components 5-12 in Figure 3 do not appear periodic since they are composed of the Fourier components B1-8 in Figure 15.

*10) The eigenvectors of a correlation matrix are orthogonal. If we normalize these eigenvectors and furthermore enforce an additional constraint that the entries of each eigenvector must be positive semidefinite, then the "synaptic weights" of different eigenvectors should not overlap. In other words, if an entry in the first eigenvector is non-zero, it ought to be zero in all other eigenvectors.*

In the constrained case, the solutions maximizing the variance subject to the non-negativity constraint are no longer eigenvectors of any matrix. Thus, the second such solution described in the section “Hierarchical networks and modules” need not be orthogonal to the first, and the issues raised by the reviewers does not arise. We have added a note on this issue in the Results section: “It is important to note though that due to the non-negativity constraint the vectors achieved this way were not orthogonal, and thus it cannot be considered a real orthogonalization process.”. See Figure 18, demonstrating that the different vectors outputted from the simulation are indeed not orthogonal.

*11) While the algorithms the authors use relax this last constraint, it is not clear how skewing or shearing a square pattern that results from PCA helps satisfy the dual constraints of non-negativity and orthogonality. This is a key point in the paper, and begs for an intuitive explanation, or at least more analysis or quantification of the degree to which the constraints are satisfied.*

We believe the above reply addresses this issue. See also Figure 18.

*12) The finding that the spacings corresponding to successive eigenvectors obey ~*
2
*also calls for an explanation. Why should this ratio of spacings best satisfy non-negativity, orthogonality, and periodicity?*

We now provide an analytical motivation for this in the new revised Methods section of the paper. In short, the change of scale occurs when “jumping” from the first optimal circle in the Fourier space to next available Fourier lattice points which can form a hexagonal shape (Figure 17 in paper).

[Editors' note: further revisions were requested prior to acceptance, as described below.]

*Reviewer #1: The authors have worked hard to provide further intuition and interpretation to their results, with much success, although there are still some points that should be clarified, see comments below. There is also a plain error that should be removed. The authors state "the distribution of grid orientations becomes uniform in the range of 0-15 degrees, with a mean (and median) at 7.5 degrees, similar to experimental data [Stensola et al., 2015]". On the contrary the experimental data shows a very non-uniform distribution clearly peaked at about 7.5 degrees, which is quite different. Also, as a circular variable limited to [0,15) – albeit one for which 16 degrees corresponds to 14 degrees rather than 1 degree – the 'mean' and 'median' they refer to are undefined if the distribution is uniform. Thus the authors should acknowledge that they do not provide an explanation for this aspect of the experimental data, and some of the text/ results on this point could be removed, most importantly in the Abstract.*

We agree that the experimental data is non-uniform, and clearly peaked around 7.5^o^. We have thus erased this point from the abstract, and removed it throughout the paper.

*At the start of describing the main result (Results, first paragraph) is not clear that the "1d mapping of the 2d activity" is caused by the trajectory of the rat moving through the place fields.*

This point is now clarified in the second paragraph of subsection “Comparing Neural-Network results to PCA).

*The next part of the description of their PCA analysis as Eigenvectors and their spatial interpretation could also be improved in terms of intuition. It would be worth mentioning that the PCA, in addition to being calculated as the Eigenvectors of the matrix of temporal covariance between inputs, can also be thought of in terms of singular value decomposition of the (appropriately normalised) input activity. I.e. the input to the network (temporal sequences of place cell activity) can be decomposed into an outer product of (spatial) weights over place cell inputs (their Eigenvectors) and temporally varying weights over these Eigenvectors. The network learns the spatial weights over place cells (the Eigenvectors) as the connections weights from the place cells, and "projection onto place cell space" is simply the firing rates of the output neuron plotted against the location of the rat.*

Thanks for this good point. This alternative interpretation using SVD has now been added to the in the Methods section.

*In the second paragraph of the subsection “Adding a non-Negativity constraint to the PCA “it is important to remind the reader that the "raw place cells activity" is not the final input to the PCA numerical methods – spatial or temporal mean normalization has to happen first.*

This reminder has now been added in the third paragraph of the section “Adding a non-Negativity constraint to the PCA”.

*It would be useful to have a surface plot showing Gridness vs. PC Density vs. PC Width. I have found it hard to replicate the Gridness distribution and wonder how general it is to the precise place cell inputs used?*

We have now added the following subpanels to Figure 12, in which we vary the ratio between the box size and the size of the input place fields. As can be seen, as long as the box and place field sizes are large enough, the Gridness is > 1 (left panel). Furthermore, the scale of the grid is not highly dependent on the size of the box (right panel).

*The reference to Figure 10 in the third paragraph of the subsection “Effect of place cell parameters on grid structure “is incorrect (I think) – not showing the dependence on the width of the place fields.*

Typo has been corrected to “Figure 12”.

*Regarding the presence of "modules" of grid cells (subsection “Modules of grid cells”), it is not clear if the interpretation is that different modules correspond to different grid scales, or different Eigenvectors (even if having the same scale). Assuming the latter, it seems that the analysis predicts that there should only be two modules, with the scale of the first module approximately 7.5x the largest place field inputs, and the second module (~*2
*smaller)*

*existing only to enable non-negative PCA solutions. This deserves discussion, given that the data show at least 5 modules, with a range of scales, the smallest and most numerous having approximately the scale of the smaller place fields found in the dorsal hippocampus (25-30cm).*

We agree. We believe that a thorough understanding of the module phenomenon will require further study. Mainly two points need to be added to a future model: (a) Looking at a non-uniform distribution of place-cell sizes, similar to reality and (b) adding the feedforward projection from grid cells to place cells, thus “closing the loop”. This point has now been added to the Discussion.

Reviewer #2: The last equation in the section “2D Solutions”: should read ϕ_1_ = ϕ_2_ = ϕ_3_n 2 π/3, n ϵ Z. e.g. ϕ_1_ = ϕ_2_ = ϕ_3_

*= 0 would be a solution. As written, ϕ_1_ = ϕ_2_ = ϕ_3_, is not the maximum. Besides, this more general case would allow solutions that are unlike the ones observed in experiments.*

This part was removed due to our revision, in response to the next comment. In any case, this was a typo (which happened due to an inaccurate change of notation from cos(k *_i_* ·(x-x_o_)) to cos(k *_i_* ·x+*ϕ_i_*). It should have been “*ϕ_i_* = k*_i_* ·x_0_, for some x_0_”.

This brings me to In the subsection “2D solutions”: "To do this [maximize the Fourier components inside the basis set,] we must maximize the minimal value of the cosine components for the basis set" I am afraid this argument escapes me, even though its role is crucial.

While I think I understand the Fourier domain argument up to this point in the text, how exactly does maximizing the minimum value in real space of the function guarantee that the Fourier components in Eq. 24 are maximized?

This argument was indeed incorrect (it is true if can we neglect the higher order harmonics in each lattice, unfortunately this cannot be done, as we verified numerically). We thank the reviewer for noting this. We changed the sub-section “2D solutions” (in section “The PCA solution with a non-negative constraint”) to correct this. First, we prove (in the appendix section “Upper bound on the constrained objective – 2D square lattice”) that the objective is bounded from above by 0.25 in any non-hexagonal 2D lattice with a single grid length (i.e., a square or 1D lattice). Note this remains true even if we consider higher order harmonics. Second, we give an example for a hexagonal lattice (found via a numerical scan) which achieves an objective value higher than 0.25. Thus, given our approximations, this shows that any optimal solution must be a hexagonal lattice

Eq. 34: why is the upper limit here ∞, when it was M in Eq. 33?

The two equations result from two different numerical procedures and assumptions. We revised the text near the two equations to explain this point:

“This form (Eq. 33) results from a parameter grid search for *M* =1, 2,3 and 4, under the assumption that *L* ⟶ ∞ and *r*ˆ *(k)* is highly peaked. […] Specifically, it has similar form (Eq. 34).”

The main point is to show that both give very similar results. In other words, that our approximate description, using the simplifying assumptions and finite *M*, is closely related to the solution without approximations. Additional details were added to the text to clarify this important point. Also, to improve focus, we relegated to the appendix some less important details from that section

(on the origin of the difference between *k*_†_ and *k*_*_).